# Arctic sea ice sensitivity to lateral melting representation in a coupled climate model

Madison M. Smith[1], Marika Holland[2], and Bonnie Light[1]

[1]Applied Physics Lab, University of Washington, Seattle, Washington, USA
[2]National Center for Atmospheric Research, Boulder, Colorado, USA

**Correspondence:** Madison Smith (mmsmith@uw.edu)

**Abstract.** The melting of sea ice floes from the edges (lateral melting) results in open water formation and subsequently increases absorption of solar shortwave energy. However, lateral melt plays a small role in the sea ice mass budget in both hemispheres in most climate models. This is likely influenced by the simple parameterization of lateral melting in sea ice models that are constrained by limited observations. Here we use a coupled climate model (CESM2.0) to assess the sensitivity of modeled sea ice state to the lateral melt parameterization in pre-industrial and $2xCO_2$ runs. The runs explore the implications of how lateral melting is parameterized, and structural changes in how it is applied. The results show that sea ice is sensitive both to the parameters determining the effective lateral melt rate, as well as the nuances in how lateral melting is applied to the ice pack. Increasing the lateral melt rate is largely compensated by decreases in the basal melt rate, but still results in a significant decrease in sea ice concentration and thickness, particularly in the marginal ice zone. Our analysis suggests that this is tied to the increased efficiency of lateral melting at forming open water during the summer melt season, which drives the majority of the ice-albedo feedback. The more seasonal Southern Hemisphere ice cover undergoes larger relative reductions in sea ice concentration and thickness for the same relative increase in lateral melt rate, likely due to the hemispheric differences in the role of the sea ice-upper ocean coupling. Additionally, increasing the lateral melt rate under a $2xCO_2$ forcing, where sea ice is thinner, results in a smaller relative change in sea ice mean state, but suggests that open water formation feedbacks are likely to steepen the decline to ice-free summer conditions. Overall, melt processes are more efficient at forming open water in thinner ice scenarios (as we are likely to see in the future), suggesting the importance of accurately representing thermodynamic evolution. Revisiting model parameterizations of lateral melting with observations will require finding new ways to represent salient physical processes.

## 1 Introduction

Sea ice in the Arctic and Antarctic undergoes strong seasonal changes. A key factor in the annual retreat of sea ice cover is the sea ice-albedo feedback. This feedback encompasses a variety of changes in ice mass and surface characteristics (such as the melt of snow and formation of melt ponds), but the largest component of this feedback is typically considered to be the loss of ice-covered area to open water area (Curry et al., 1995). In addition to formation of open water from thermodynamic ice melt,

dynamics can play a significant role in the formation of open water in some regions as a result of advection and ridging (Rigor
et al., 2002).

Throughout the summer, the absorption of solar shortwave radiation promotes ice melt. Absorption at the ice surface causes surface melt, and absorption by the ocean increases ocean heat. Heat in the upper ocean promotes basal and lateral sea ice melt. All three of these melt processes (surface, basal, and lateral) contribute to open water formation by reducing ice volume with varying effects on ice thickness and area. Vertical melt processes (surface and basal) can only form open water once the ice is thin, while lateral melt can directly form open water area regardless of ice thickness.

Coupled climate models are a primary tool for assessing changes in sea ice cover in past and future climates, and the ice-albedo feedback is key to being able to realistically simulate changes (e.g., Holland et al., 2006b). The role of this feedback depends on a variety of model parameterizations and choices. Clearly, the formation of open water is dependent on the representation of melt processes. Of particular relevance here, seven of the 15 Climate Model Intercomparison Project Phase 6 (CMIP6) models reviewed by Keen et al. (2021) had no explicit representation of lateral melt. There is wide model variation in the partitioning of mass flux between melt processes, but the multi-model mean allocates only 4% to lateral melt while the vertical melt processes account for a combined 77%, with great spread in the relative ratio of surface to basal melt (Keen et al., 2021).

Additionally, the implementation of sea ice melt processes in coupled climate models depends on other choices in the sea ice and ocean models. In particular, the representation of the sea ice cover using an ice thickness distribution results in a stronger albedo feedback because the better resolution of thin ice enhances thermodynamic ice loss (Holland et al., 2006a). As a result, climate models including a sea ice component with sub-grid scale ice thickness distributions have best matched observations of sea ice extent over the satellite era (Stroeve et al., 2007). Bitz et al. (2001) states that "resolving thin ice [using the ice thickness distribution] eliminates the need for partitioning an unrealistically high fraction of heat flux toward lateral melt", indicating the importance of the ice thickness distribution in simulating melt rates. Lateral melt can have an important role in driving feedbacks in a manner similar to the thickness.

Relatively little work has been done to understand the role of lateral melt in sea ice evolution, with the foundation for much of what is known coming from a few observations made in the 1980s. Perovich (1983) used a dataset of lateral melt at a static lead in the Canadian Arctic to estimate empirical parameters for the formulation of Josberger and Martin (1981), relating the temperature of the water above freezing to the melt rate. Later, Maykut and Perovich (1987) suggested another formulation for lateral melting, which included wind friction velocity ($u_*$), using a somewhat larger set of observations from the 1983-1984 Marginal Ice Zone Experiment (MIZEX). However, the inclusion of wind friction velocity did not result in significant improvement in fit with observations and is more difficult to constrain, and so was not largely adopted.

Both observations and modeling studies have suggested that it is important to resolve the partitioning of solar energy absorbed in leads and the upper ocean between lateral and basal melting. Maykut and Perovich (1987) used observations from the Greenland Sea to suggest that lead width, orientation, and current velocities drive the relative degree of lateral melting. Steele (1992) used a simple model to assess partitioning of lateral and basal melting in the Arctic given by empirical parameterizations. Steele suggests that under the typical Arctic summer forcing, lateral melting is significant only for floes with average

diameter on the order of 30 m or less, while another study suggests that lateral melt is sensitive to floe size with diameters 100 m or less (Tsamados et al., 2015). Hunke (2014) found that in a coupled climate model, lateral melting may be important, despite being a small term in the mass budget, because of the sensitivity of sea ice to thermodynamic processes, especially in thin ice categories. Model experiments showed that although lateral and basal melt are both driven by heat in the ocean, they may respond to model changes in different ways. Skyllingstad et al. (2005) developed an large-eddy simulation model for leads which showed that freshwater stratification in leads plays a major role in controlling lateral melting. It is clear that further field data are required to illuminate the relationships of physical controls with lateral and basal melt rates, as existing empirical relationships (Perovich, 1983) have large uncertainties as a result of limited observations.

The lateral melt rate has been suggested to be particularly sensitive to floe sizes (Steele, 1992; Roach et al., 2018, 2019; Boutin et al., 2020; Bateson et al., 2020). With recent model developments to include a floe size distribution (e.g., Zhang et al., 2015; Roach et al., 2018, 2019; Boutin et al., 2020, 2021; Bateson et al., 2020), there has been renewed interest in the modeling community to properly describe lateral melt. However, all sea ice models incorporating floe size still utilize the empirical relationship for lateral melting introduced by Perovich (1983). There is a notable disconnect in the evolution of models to include more complex and realistic physics and the progress of our physical understanding of lateral processes that drive these changes. Roach et al. (2018, 2019) and Bateson et al. (2020) explored the sensitivity of sea ice to higher lateral melting by incorporating a floe size distribution with smaller floe sizes. They found that increased lateral melting was largely compensated for by decreased basal melting in standalone sea ice models, but the reduction in basal melt was smaller than the increase in lateral melt in a coupled sea ice-ocean setup (Roach et al., 2019). Even further differences might be expected in a model with a coupled atmosphere that allows feedbacks related to the formation of open water.

Previous studies have explored the importance of open water for sea ice mean state by using what was defined by Holland et al. (2006b) as the open water formation efficiency: the area of open water formed in a region as a result of a unit reduction in sea ice volume. The mean sea ice thickness and volume are strongly related to the total summer open water formation efficiency in models, as the open water formation is important for sea ice volume evolution (Holland et al., 2006b; Massonnet et al., 2018). The open water formation efficiency has since been used to understand the trajectory of the Arctic towards an ice-free summer, indicating the importance of melt processes on the ice-albedo feedback in capturing sea ice response to forcings (i.e., Massonnet et al., 2018; Lindsay et al., 2009; Merryfield et al., 2008). While the efficiency of different melt processes at forming open water has not been explored, we expect that the relative magnitude will play a role in determining the resulting sea ice state.

In the present paper, we modify the lateral melt parameterization in a coupled global climate model, CESM2.0. The main objective of this study is to investigate to what extent lateral melting can affect the simulation of sea ice in a coupled climate model as a result of factors driving sea ice change associated with open water formation, including the ice-albedo feedback.

 ## 2   Model and experimental design

### 2.1   CESM2.0

Sensitivity tests were completed with CESM2.0 (Danabasoglu et al., 2020) using a constant pre-industrial forcing, over a global model domain with nominal horizontal resolution of 1°. Runs were branched from year 880 of the CESM2.0 CMIP6 preindustrial control run (Danabasoglu et al., 2020), providing fully initialized states for all model components. CMIP6 preindustrial
control runs are forced by interannually invariant atmospheric conditions appropriate for the year 1850.

The model runs completed here have fully coupled atmosphere, sea ice, and land models, and a simplified slab ocean model (SOM). The SOM replaces the standard full-depth ocean model with a surface ocean mixed layer governed by the equation (Eq. (1) from Bitz et al., 2012):

$$\rho_o c_p h \frac{\delta \text{SST}}{\delta t} = F_{net} + Q_{flx} \tag{1}$$

where $\rho_o$ is the density of seawater, $c_p$ is the ocean heat capacity, $h$ and $Q_{flx}$ are respectively the mixed layer depth (MLD) and ocean heat flux convergence (associated with advection and mixing) obtained from a fully coupled CESM2.0 run, and $F_{net}$ is the net heat flux to the ocean. The prescribed MLD $h$ varies spatially based on climatological conditions simulated by CESM2.0 coupled runs with an active ocean component. However, it is constant over time (i.e., doesn't vary seasonally or inter-annually), and has a minimum depth of 10 m. The prescribed ocean heat flux $Q_{flx}$ varies both spatially and seasonally.
The mixed layer temperature (here defined as equivalent to the sea surface temperature, SST) evolves with surface heat fluxes determined by the coupled climate model; thus ice-albedo feedbacks are permitted. A result of the prescribed MLD and $Q_{flx}$ is that feedbacks associated with ocean dynamics and mixing are not present, and there is no ability for variability in ocean dynamics to drive changes in ocean heat content.

The use of the SOM requires significantly less computational time as a result of not running a full ocean model, and because
it allows the model to converge much faster - around 30 years vs. 100s of years for the fully coupled model. Nonetheless, runs with a SOM reproduce the climate of the fully coupled model quite well (Bitz et al., 2012); it has been particularly used to assess climate sensitivity (Bacmeister et al., 2020; Gettelman et al., 2019) as the fully coupled atmosphere permits climate feedbacks, even though feedbacks associated with ocean dynamics are inactive. Thus, implementing the SOM allows us to run multiple sensitivity tests in a coupled climate that would be prohibitive with the fully coupled dynamic ocean. Roach et al.
(2019) used the SOM for experiments focused on sea ice floe size, justified by qualitatively similar results between the slab ocean model and a fully-coupled dynamic ocean model. Changing lateral melt is mostly expected to affect the upper ocean thermodynamically by changing shortwave absorption, and these interactions are included in this model setup.

The sea ice model used is CICE version 5.1.2 (Hunke et al., 2015), which uses mushy-layer thermodynamics by default (Turner and Hunke, 2015; Bailey et al., 2020). Other key features include the elastic-viscous-plastic rheology and sub-grid
ice thickness distribution, with a default of five categories, as standard in prior versions of the CICE model (Fig. 1). Sea ice simulated by CESM2.0 over the historical period has reasonable mean state and variability in both hemispheres (DeRepentigny et al., 2020). The version used here uses tuned albedos for a more realistic simulation of ice thickness (Kay et al., in review).

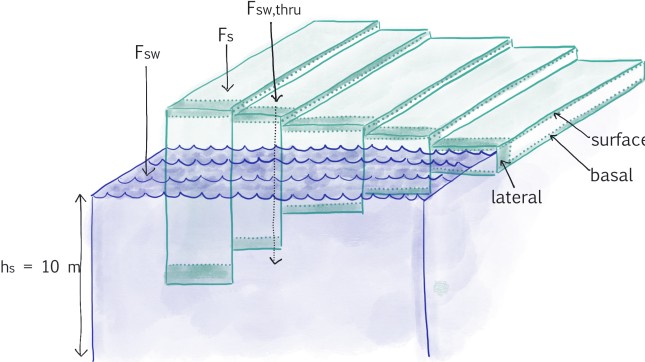

**Figure 1.** Schematic of key components of a model grid cell in the CICE sea ice model, including the five-category ice thickness distribution, and important fluxes and melt terms. Based on schematics from Notz and Bitz (2017) and Petty et al. (2014).

Specifically, the albedo of snow on sea ice was increased by decreasing the snow grain radius (with an increase of the parameter $r_{snw}$ from 1.25 to 1.5), and the temperature at which snow grain growth is permitted to occur (due to melting conditions) was increased by 0.5 °C from 1.0 to 1.5 °C.

Each run was at least 60 years long, and averaging was typically done over the last 25 years (simulation years 35-60) after the system has equilibrated and to minimize the contribution from internal variability. Monthly-averaged outputs are used to examine changes in mean state of the sea ice for computation efficiency and for better comparison with other studies (which typically use monthly averages), while daily averages are required to examine efficiency of melt processes.

### 2.1.1   Representation of the ice thickness distribution

The CICE model includes an ice thickness distribution, which is common across most modern global sea ice models (Keen et al., 2021). Sea ice is discretized into a set number of categories, which occupy an evolving fraction of the grid cell. Sea ice volume and area are transferred between categories as a result of sea ice melt and growth, as well as dynamic processes. Lipscomb (2001) introduced a linear remapping scheme to transfer ice between categories, which has faster convergence and is less diffusive than prior schemes.

Based on the results of Bitz et al. (2001), a majority of global climate models with a sea ice component use five thickness categories as the default (Keen et al., 2021), where all are assumed equally in contact with open water (Fig. 1). Runs in a coupled climate model showed that this number struck a balance of capturing the features of runs with higher numbers of categories, while still keeping computational load low. However, there are a number of different schemes for determining the

boundaries of the discretized ice thickness categories. The original discretization scheme, used here, follows the definition of category boundaries from Eq. (22) in Lipscomb (2001) which gives more widely spaced boundaries for thicker ice. The higher resolution for thin ice is beneficial to minimize diffusion, as well as to better resolve the evolving mass budget. The mass budget has a non-linear sensitivity to ice thickness because of the inverse relationship with congelation growth, where thick ice grows slowly and thin ice grows rapidly (e.g., Massonnet et al., 2019).

### 2.1.2 Parameterization of lateral melting


The lateral melt rate is determined based on an empirical relationship with the difference of the SST from freezing ($\Delta T$) giving a uniform melt rate around the perimeter of the floe:

$$w_{lat} = m_1 \Delta T^{m_2} \tag{2}$$

where the constants are based on empirical estimates from a single static lead in the Canadian Arctic by Perovich (1983),
$m_1 = 1.6 \times 10^{-6}$ (units m s$^{-1}$ deg$^{-m_2}$), and $m_2 = 1.36$ (unitless).

This is then used to calculate the change in ice concentration ($a$) for each thickness category as a result of lateral melt

$$\left(\frac{da}{dt}\right)_{lat,n} = \frac{w_{lat} * dt * \pi}{\alpha * D} \tag{3}$$

over a given time-step, $dt$ (Steele, 1992). Effective floe diameter ($D$) is a constant value of 300 m by default that does not evolve with melting. $\alpha$ is a floe shape parameter representing the non-circularity of floes set at a default of 0.66 (Rothrock and
Thorndike, 1984). Note that the change in area in each category over a timestep $\left(\frac{da}{dt}\right)_{lat,n}$ is referred to as $r_{side}$ in the model.

This lateral melting parameterization is commonly used throughout coupled climate models. For example, six of the eleven CMIP6 models examined by Keen et al. (2021) include a lateral melt parameterization, and all of these use the sea ice model CICE with the same lateral melt parameterization described here. We note again here that while there are recent model developments to include an evolving floe size distribution based on coupled processes (e.g., Zhang et al., 2015; Roach et al., 2018,
2019; Boutin et al., 2020, 2021; Bateson et al., 2020), most models do not currently have the capability to have a variable floe size or include the necessary couplings (such as surface waves).

Finally, the lateral melt rate, $(dV/dt)_{lat}$, is calculated as

$$\left(\frac{dV}{dt}\right)_{lat} = \sum_n V_{ice,n} \left(\frac{da}{dt}\right)_{lat,n} \tag{4}$$

where $V_{ice,n}$ is the volume of ice in each thickness category, $n$.
The temperature of the surface ocean above freezing (i.e., $\Delta T$) is also used to determine the basal turbulent heat exchange driving basal melt, in addition to lateral melt. The sum of ocean flux terms is not allowed to exceed the heat content in the ocean surface layer. If the sum of lateral and basal heat flux represents a larger flux than that available, the lateral and basal heat fluxes are both reduced by the scalar factor necessary such that all ocean heat content is lost to the ice.

For a selected part of the historical period (1960-1989), CESM2 predicted 0.4 Gt$\times 10^3$ mass loss per year associated with
lateral melting, which is about 4.2-4.8% of the total mass loss per year associated with vertical melting (with the range representing configurations with different atmospheric models; Keen et al., 2021). With the pre-industrial forcing and configuration

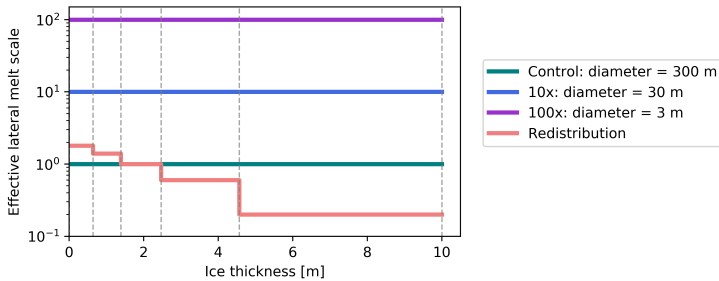

**Figure 2.** Summary of lateral melt rates for all sensitivity runs. Vertical dashed grey lines indicates bounds of the 5 ice thickness distribution categories. Solid lines show the effective lateral melt rate scale, relative to a value of 1 for the default parameterization. Decreasing the floe size $D$ and increasing the scale $r_n$ have the same effect, such that the effective lateral melt rate scale can be summarized as $\frac{r_n}{D/300}$, where 300 is default floe diameter.

of CESM2 used in this study, the average annual volume loss from lateral melting is similarly about 4.5% of the volume loss associated with vertical melting (Fig. 5).

## 2.2 Sensitivity experiments

The control run using default parameterizations of lateral melting as described above (i.e., floe diameter = 300 m) was run using a constant pre-industrial forcing for sixty years, and three additional experiments of 60-year duration were performed. To test the impact of increased lateral melting, we completed two runs where the constant floe diameter $D$ was decreased to 30 m and 3 m, where floe size is used as a convenient approximation for lateral melt rate in the context of this study. These decreases in floe size effectively increase the lateral melt rate by 10x and 100x, respectively. Observations of floe size distribution are limited, and do not have sufficient spatial or temporal coverage to determine what floe size is most representative for ice-covered regions; the most complete coverage is provided by satellite products such as CryoSat-2 (Horvat et al., 2019), but the relevance for lateral melt is significantly limited by the footprint of 300 m. Notably, our sensitivity simulations are not designed to assess the influence of a more realistic floe diameter, but instead to investigate the cumulative influence of uncertainties in the magnitude of lateral melting and to explore the physical processes that govern the sea ice response. As such, changes to the floe size are used as a catchall for factors impacting the sensitivity of the lateral melt rate. Other parameters controlling the rate of lateral melting as a function of temperature difference, $m_1$ and $m_2$, were kept the same, but it is noted that the effect of increasing $m_1$ is the same as decreasing the diameter $D$ (Fig. 2) such that the 10x and 100x sensitivity runs can alternatively be thought of as $m_1 = 1.6 \times 10^{-5}$ and $m_1 = 1.6 \times 10^{-4}$, respectively. There is substantial uncertainty in the default $m_1$ and $m_2$ parameter values, which were derived from a single set of observations (Perovich, 1983). Therefore, these large perturbations are justified by the uncertainties in the functional form of this parameterization and allow us to examine how they impact the sea ice and climate system. As the lateral melt already comprises a small fraction of the mass budget in the control run (Fig. 5), sensitivity runs with decreased lateral melting were not completed.

For the control and increased lateral melt rate (10x and 100x) runs, Eq. (4) suggests that lateral melt occurs equally in all thickness categories such that all decrease by the same percent each time step. However, we can imagine a number of physical reasons why lateral melt rate might be unequally distributed across the ice thickness categories, including: (i) the possibility that the upper ocean is stratified as a result of heating during the melt season (i.e., Holland, 2003), (ii) the potential for a relationship between ice thickness and floe size, (iii) cases where the open water area (leads) that result in lateral melting are unequally distributed across the ice thicknesses. While we can not yet explicitly test the impact of these processes in the model, we can explore the impact of the non-uniform lateral melt they may result in. Although there are not yet large-scale observations of floe size or lead distribution with respect to ice thickness, we hypothesize that thinner ice will have smaller floe sizes on average, and that open water is more likely to be adjacent to thinner ice categories due to the lower ice strength. Therefore, we perform an additional sensitivity test of the model distribution of lateral melting by increasing the melt rate in thin classes and decreasing it in thick classes. Eq. (4) is altered to include an ITD re-distribution factor ($r_n$) over the $n$ ice categories:

$$\left(\frac{dV}{dt}\right)_{lat} = \sum_n \left(\frac{dV}{dt}\right)_{lat,n} = \sum_n V_{ice,n} \left(\frac{da}{dt}\right)_{lat,n} r_n \tag{5}$$

with $r_n = [1.8, 1.4, 1.0, 0.6, 0.2]$. These values were distributed around 1 with the aim of keeping the total lateral melt volume approximately the same, such that the effect of the redistribution can be primarily observed. The focus here was on making simple changes to understand the impact of the limitations in the lateral melting parameterization itself, rather than on prescribing what an appropriate distribution of lateral melting should be. Note that the category-dependent melt values were only saved for the last 5 years of this run, and so average open water formation efficiency values (see Sect. 2.3) are only over 5 years, compared to 25 years for other runs. The results are qualitatively unchanged by averaging over 5 years for all runs (not shown).

The schematic in Fig. 2 summarizes the effective changes to lateral melt rate in each of these runs compared to the control. The changes to the effective lateral melt rate scale are larger for the 10x and 100x runs (blue and purple, respectively) than for the redistribution run (light red), but remain evenly applied across all thickness distribution categories (delineated by the grey dashed lines).

## 2.3 Open water formation efficiency of melt processes

We build off the definitions of open water formation efficiency (OWFE) used by Holland et al. (2006b) and Massonnet et al. (2018) to define OWFE for individual melt processes as a ratio of the associated area of open water formed ($da/dt$) and the volume of sea ice melt ($dV/dt$) (e.g., the area of open water formed per unit volume melted):

$$\text{OWFE} = \frac{da/dt}{dV/dt} \tag{6}$$

This permits the examination of the contribution of melt processes to the albedo feedback, regardless of the total volume of melt that they currently result in within the model. We look only at grid cells from daily outputs with non-zero melt rates and

zero total growth rates in order to isolate the influence of melt processes. The OWFE of lateral melt is specifically defined as:

$$\text{OWFE}_{lat} = \frac{(da/dt)_{lat}}{(dV/dt)_{lat}} \tag{7}$$

In general, the calculation of change in concentration from lateral melt is calculated as:

$$\sum_n \left(\frac{da}{dt}\right)_{lat,n} = \sum_n \left(\frac{dV}{dt}\right)_{lat,n} \frac{1}{V_{ice,n} r_n}. \tag{8}$$

With the default model configurations, where lateral melt as defined in Eq. (4) is applied to all thickness categories equally, this simplifies to:

$$\text{OWFE}_{lat} = \frac{1}{V_{ice}} \tag{9}$$

Thus, the area of open water formed from lateral melting is directly proportional to the average ice thickness in the grid cell in the control and sensitivity runs with increased lateral melt rate. For the run with lateral melt redistribution, the OWFE of lateral melt will additionally include the redistribution factor:

$$\text{OWFE}_{lat} = \frac{\sum_n [(dV/dt)_{lat,n} \cdot V_{ice,n}^{-1} r_n^{-1}]}{(dV/dt)_{lat}} \tag{10}$$

Vertical melt is a result of both basal and surface melt. The efficiency of vertical melt processes ($\text{OWFE}_{vert}$) is calculated using the difference between the total change in concentration due to thermodynamic processes (output daily from the model as $(da/dt)_{thermo}$) and the change in concentration due to lateral melt:

$$\left(\frac{da}{dt}\right)_{vert} = \left(\frac{da}{dt}\right)_{thermo} - \left(\frac{da}{dt}\right)_{lat} \tag{11}$$

The open water formation efficiency of vertical processes is then:

$$\text{OWFE}_{vert} = \frac{(da/dt)_{vert}}{(dV/dt)_{basal} + (dV/dt)_{top}} \tag{12}$$

The schematic in Fig. 1 helps to demonstrate how the OWFE of melt processes is related to the ice thickness distribution. In the standard lateral melting parameterization, the lateral melting rate is applied to all categories equally. While sea ice melts equally in all categories, thin ice categories will form open water most rapidly such that the OWFE of lateral melt should be directly tied to the average thickness. In contrast, vertical melt rates in all runs are determined by the balance of fluxes and so are unevenly distributed across the categories. However, open water formation from vertical melt processes is driven by ice in the thinnest category, as it will result in open water formation only when a portion of ice completely melts through.

## 3 Results

### 3.1 Changes in ice concentration and thickness

Figure 3 shows the seasonal cycle of sea ice area, volume, average thickness, average fractional coverage (calculated as the ratio of sea ice area to extent), and thick ice area in the Northern Hemisphere with a constant pre-industrial forcing. As is

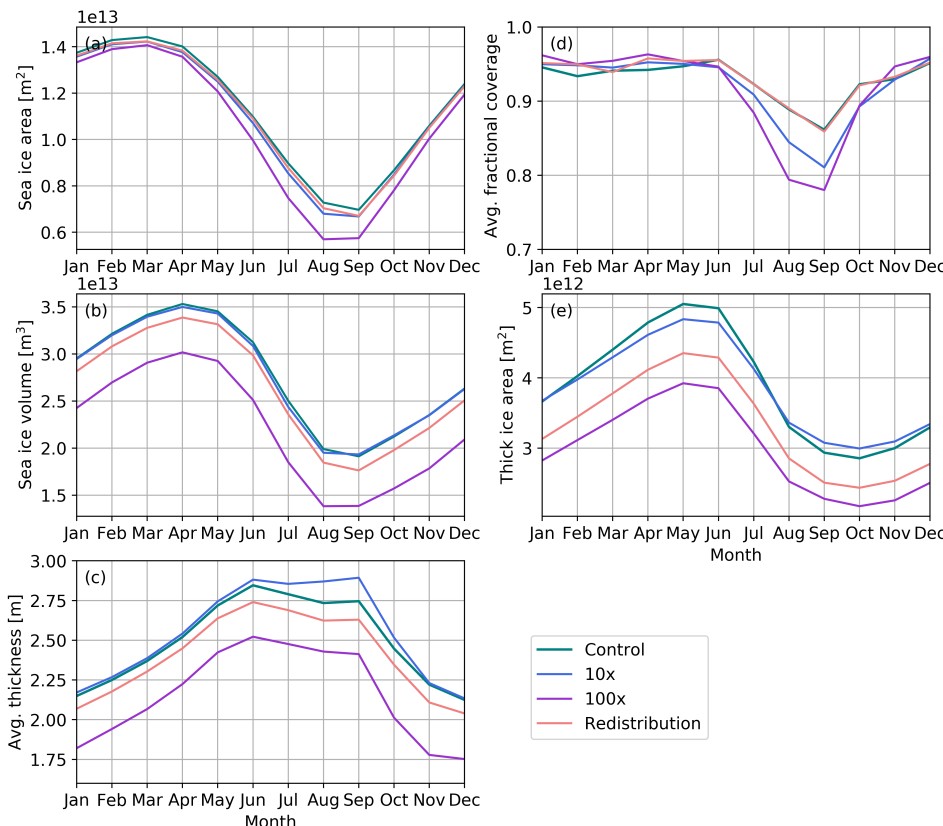

**Figure 3.** Seasonal cycle of Northern Hemisphere (a) sea ice area (m$^2$), (b) sea ice volume (m$^3$), (c) average sea ice thickness (m), (d) average fraction of sea ice covered area, calculated as ratio of sea ice area to sea ice extent, and (e) area with thick ice (greater than 2.5 m) for all sensitivity runs with pre-Industrial forcing - control run (teal), 10x lateral melt run (blue), 100x lateral melt run (purple), and redistributed lateral melt (light red). Results shown are averages over the last 5 run years for the redistributed lateral melt run, and the last 25 years for all others.

convention, hemispheric sea ice area and volume are the sums of individual grid-cell areas and volumes, respectively, and extent is the sum of grid cell areas where concentration exceeds 15%. The control run shows the maxima in area is in March and volume is in April, while the minima for both occur in September.

    The most notable effect on sea ice state in sensitivity runs is that increasing the lateral melt rate reduces sea ice volume (Fig.
3b). However, there is a minor response to the 10x increase in lateral melting, while the 100x increase drastically reduces sea ice volume (by approximately $5 \times 10^{12}$ m$^3$ during all months), and moderately reduces sea ice area (Fig. 3a). In fact, the average thickness is somewhat higher July to October in the 10x run compared to the control, while it is significantly reduced in the 100x run (Fig. 3c). Redistribution of lateral melt to thin categories reduces the sea ice volume across all months. Contrary to intuition, substantial increases in the lateral melt rate do not necessarily result in reductions of sea ice area and volume of a
comparable magnitude.

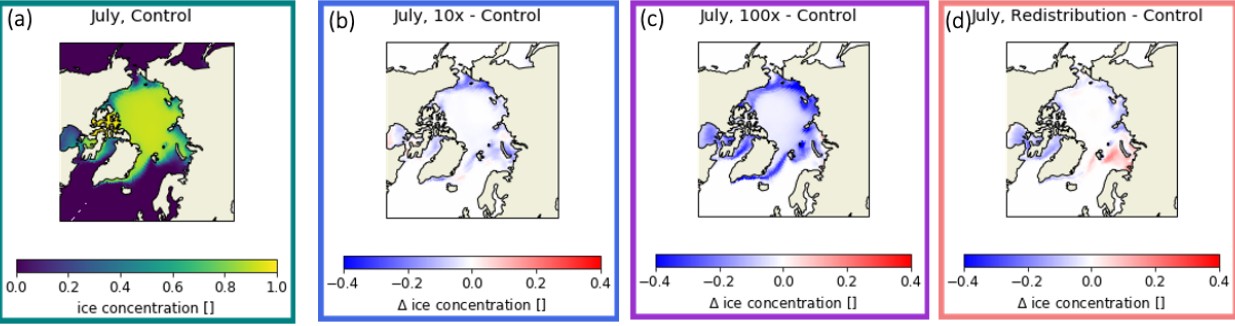

**Figure 4.** Maps of July Northern Hemisphere sea ice concentration (a) in the control run, and relative change in the (b) 10x, (c) 100x, and (d) redistributed lateral melt runs.

The ratio of sea ice area to sea ice extent is approximately equal to the average fractional coverage (Fig. 3d). All runs show values between 0.9 and 1 from November to June, but are lower during the summer melt season, suggesting more open water in the ice-covered areas during this time. While the control and redistribution runs reach a minimum of about 0.86, both the 10x and 100x runs reach significantly lower, with minima of 0.81 and 0.78 respectively. This suggests more potential for atmosphere-ocean interactions in ice-covered areas when the lateral melt rate is higher. Increasing the lateral melt rate primarily reduces the sea ice concentration around the margins (Fig. 4b-c), where the sea ice is relatively thin and has low concentration during the summer (Fig. 4a). For example, there is a substantial decline ($\sim$20-30% in July) in Hudson Bay in the 100x run compared to the control (Fig. 4c). Here, the ice is relatively thin and increasing the efficiency of formation of open water can contribute to an albedo feedback. The changes in spatial patterns with the redistribution of lateral melt (Fig. 4d) results from the patterns in ice thickness distribution, where the relatively high proportion of thick ice to thin ice in summer in the Barents Sea results in an increase in concentration.

## 3.2 Changes in mass budget

The average volume of sea ice grown and melted annually is comparable across all runs, as shown by mass budgets in Fig. 5. This agrees with the consistent amplitude of the sea ice volume seasonal cycle seen in Fig. 3b. This is primarily because the increase in total lateral melting (when lateral melt rate is increased) is largely compensated by a decrease in basal melting, such that the change in total sea ice melted is small. Both lateral and basal melting are driven by heat in the slab ocean model. As there is a finite amount of heat available in this layer, increasing the heat used for lateral melting reduces the heat available for basal melting. The change in the sea ice area also contributes to the observed basal melt decrease. Following the analysis by Bateson et al. (2020, Fig. 5), we find a partial contribution from loss of ice area; for example, in July it accounts for 33% in the 10x run and 48% in the 100x run, compared to the nearly complete attribution in their standalone sea ice model runs. More rapid lateral melt increases the total melt rate early in the season, when more heat is available, but decreases it later in the summer as more heat has already been used for melting and solar insolation is lower. We note that although the total

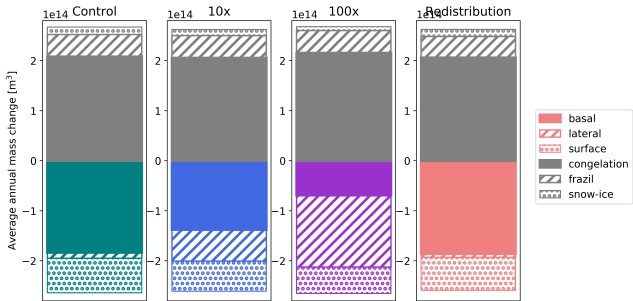

**Figure 5.** Annual Northern Hemisphere mass budgets for all sensitivity runs (from left to right: control, 10x lateral melt, 100x lateral melt, redistributed lateral melt). Grey areas represent volume of growth terms - congelation (solid), frazil (hashed), and snow-ice (dotted) - and colored areas represent volumes of melt terms - basal (solid), lateral (hashed), and surface (dotted). Results shown are averages over the last 5 run years for the redistributed lateral melt run, and the last 25 years for all others.

melt remains near constant because of the compensating effect of basal melt, the melt is distributed heterogeneously across thicknesses by design and likely leads to the resulting changes in sea ice mean state, as will be discussed further in Sect. 3.3.

Although there is a similar magnitude change in the volume of lateral melt from the control run to the 10x and 100x sensitivity runs (Fig. 5), there is a significantly more drastic decrease in sea ice area and volume in the 100x lateral melt run compared to the 10x lateral melt run (Fig. 3). This suggests that the feedbacks associated with increased lateral melting are initially small, such that the increase associated with the 10x run results in little change in total sea ice. As the lateral melt rate becomes much larger, there is more open water formed in ice covered areas over the melt season (Fig. 3c) and the feedbacks

are significant enough to result in substantial changes in sea ice mean state. This likely includes the ice-albedo feedback, as well as other processes related to dynamic and thermodynamic changes in the ice.

The portion of the mass budget accounted for by lateral melting is decreased in the lateral melt redistribution run, despite the intention to keep the total lateral melt approximately the same. This is likely a result of the adjustments to the lateral melt parameterization (Fig. 2) and the unequal distribution of sea ice between categories; the redistribution results in more rapid

melt of thin ice, which results in less thin ice to melt, and there is less lateral melting in the thick ice that remains. The seasonal cycle of the mass budget (not shown) shows that total melt is in fact increased early in the melt season, but approximately unchanged in mid-late summer.

## 3.3   Open water formation efficiencies

Figure 6 examines the total annual open water formation efficiency, as defined by Eq. (6), as well as the efficiency specifically

of lateral and vertical melt processes, as defined by Eqs. (10) and (12). In the control run, the total open water formation efficiency is on average 0.35 m$^2$/m$^3$, suggesting an average of 0.35 m$^2$ open water formed for every cubic meter of ice melted, with similar values of 0.37 and 0.34 m$^2$/m$^3$ for specifically lateral and vertical melt processes.

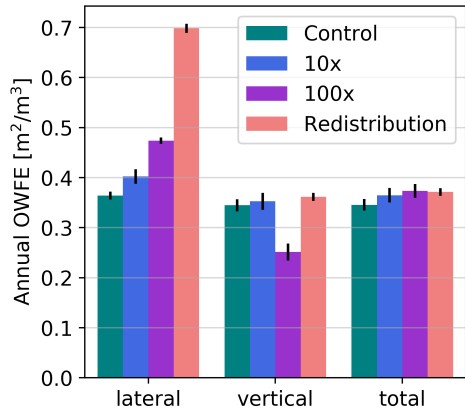

**Figure 6.** Open water formation efficiency (OWFE) for lateral melt, vertical melt processes, and as annual total in the Northern Hemisphere for all sensitivity runs with pre-industrial forcing. Colored bars show the 25-year mean, with control run in teal, 10x lateral melt run in blue, 100x lateral melt run in purple, and redistributed lateral melt in light red. Black bars denote one standard deviation. Results shown are averages over the last 5 run years for the redistributed lateral melt run, and the last 25 years for all others.

When the lateral melt rate is increased, it not only comprises a larger portion of the total ice volume melted (Fig. 5), but also forms open water more efficiently for the same volume melted (Fig. 6). The OWFE of lateral melting is more drastically

increased for the 100x run, to 0.47 m$^2$/m$^3$. As the lateral melt OWFE is closely tied to the average sea ice thickness, it is clear to see that this is a result of the thinner ice and reduction in thick ice area (Fig. 3c,e). However, it is less clear why there is still an increase in lateral OWFE for the 10x run, where average sea ice thickness is actually higher. Here, the importance of thick ice in different melt processes is underscored. The thick ice area in the 10x lateral melt run is lower than in the control run, especially during the peak of the melt season (Fig. 3e). This results in an increase in the fraction of moderate thickness ice, in

the range at which lateral melting is relatively efficient at forming open water.

The OWFE of vertical melting is significantly reduced to 0.25 m$^2$/m$^3$ for the 100x run because lateral melting more efficiently removes the thin ice. Vertical melting can only form open water from the thinnest ice categories, while lateral melting can form open water in any ice category while heat remains in the ocean. Interestingly, the vertical melting efficiency only decreases in the 100x sensitivity run. There is essentially no change in fractional or absolute area of the thinnest ice category

in 10x run, while it is significantly reduced throughout the summer melt season for the 100x run (not shown). Less ice in the thinnest thickness category means that there is less opportunity for vertical melt processes to form open water.

Interestingly, the total OWFE remains approximately constant across all sensitivity runs. This is most surprising for the 100x lateral melt run, where we may expect a higher total OWFE as a result of thinner ice on average. Massonnet et al. (2018) showed that across a range of climate models, runs with less ice volume have higher OWFE. (Though, it is worth noting that

their definition of OWFE varies from that used here in that it is defined as the linear fit between annual summer area loss and volume loss, such that it may be more related to inter-annual variability.) This suggests that for a single model, the mean state of the sea ice is more a result of the distribution of thermodynamic processes, rather than just the total OWFE. Altering the

lateral melt parameterization changes the open water formation in a way that is not fully reflected in the average thickness or sea ice mass budget.

Redistributing lateral melt towards thin categories results in large increases in OWFE of lateral melting as thin ice is more rapidly converted to open water (Fig. 6), but the vertical and total OWFE remain essentially unchanged. Both of these are a result of changes in the distribution of ice thickness. Basal melt transitions sea ice between categories, while lateral melt does not, such that an unintended consequence of reducing lateral melt in the thick categories is increasing basal melt in these categories. This results in overall less thick ice (Fig. 3e). So, the lateral OWFE increases in the redistribution sensitivity run not only because the alterations to the parameterization form open water more efficiently, but also because there is thinner ice on average, which inherently forms open water from lateral melting more efficiently. However, changes to average thickness are not substantial enough to impact the open water formation efficiency of vertical melt processes.

### 3.3.1 Timing of OWFE

As the solar shortwave energy reaching the surface in the Arctic peaks in June, the distribution of open water formation over time has an impact on the sea ice state. More efficient open water formation early in the summer is likely to result in a stronger ice-albedo feedback. In the control run, the OWFE of vertical melt processes generally peaks early in the summer, when ice is on average thinner due to the inclusion of thin first-year ice in the marginal ice zone (Fig. 7). The OWFE of lateral melt remains high throughout the melt season, and is higher than vertical melt after June 1.

The changes in basin-averaged sea ice thickness in Fig. 7b delineate two melt regimes that correspond to the temporal importance of melt processes. From April until early June, the average thickness over the entire Northern Hemisphere ice pack is increasing. This indicates that mass loss is dominated by the melt of thin ice. In mid-June, the average thickness begins decreasing, as little thin ice remains and primarily thick ice is being melted. (The decrease in average thickness in the latter half of September is a result of the beginning of new thin ice formation.) These regimes correspond with changes in OWFE; vertical melt OWFE is high primarily when melting of thin ice dominates. Lateral melt OWFE remains high when melting is primarily thick ice, all the way through until September. It is worth noting here that mass loss from both vertical and lateral melt have a similar temporal distribution, and vertical melting accounts for significantly more volume than lateral melting in the control run (Fig. 5). This result simply indicates that lateral melting that occurs during June-September results in proportionally more open water formation than does the vertical melting.

Only the 100x run is shown in Fig. 7, for simplicity, but the shapes are generally similar for all sensitivity runs with shifted magnitudes. Higher lateral melt rates result in lateral melt having higher OWFE throughout the entire melt season. The OWFE of vertical melt is lower particularly from June onwards, when the mass loss is dominated by the melt of thicker ice.

### 3.4 Sensitivity in the Southern Hemisphere

Here, we have focused on the sensitivity of modeled Arctic sea ice thickness and volume to lateral melting parameterizations. The majority of observations of lateral melting have been made in the Arctic, where the sea ice area and volume are under-

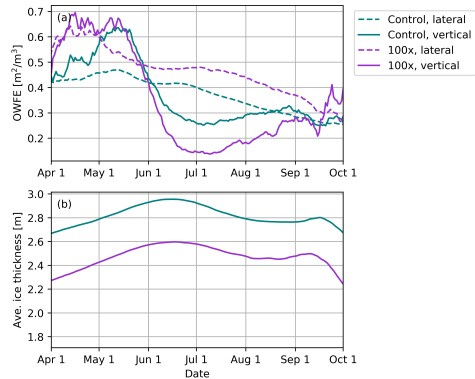

**Figure 7.** Daily (a) open water formation efficiency (OWFE) of lateral and vertical melt and (b) average sea ice thickness over the Northern Hemisphere over the melt season. Results are shown for control and 100x lateral melt runs only (teal and purple, respectively), and are averages over the last 25 run years.

going the most drastic changes. We expect the sensitivity to melt parameterizations to be somewhat different in the Southern Hemisphere, where sea ice is typically thinner and more seasonal, and there is more ocean heat availability.

Figure 8 shows the seasonal cycle of sea ice area and volume in the Southern Hemisphere. Sea ice area and volume are significantly lower year-round for all sensitivity runs. In particular, the area and volume of sea ice is approximately 50% lower during the summer months in the 100x lateral melt rate run compared to the control. The relative reductions in the 10x and 100x runs indicate a more linear response of sea ice to the lateral melt rate compared to the Arctic. Here, the 10x lateral melt and redistribution runs result in nearly the same reduction in sea ice area and volume. The redistribution of lateral melt to thin ice categories has an even larger impact than in the Arctic, likely due to the larger proportion of thin ice. It is notable that redistribution results in a comparable decrease in ice to the 10x run, despite negligible change in the lateral melt (Fig. 10). The reduction in Southern Hemisphere summer sea ice concentration is well-distributed across the ice-covered area (Fig. 9), in comparison to the changes in the Northern Hemisphere which were focused around the marginal ice zone.

As in the Arctic, the increase in total lateral melt in the Southern Hemisphere sea ice results in a decrease in total basal melt (Fig. 10). However, the total melt in the mass budgets here are lower when lateral melt is increased and there is less ice (Fig. 8b), whereas the total melt volume remained comparable across runs in the Arctic (Fig. 3b). In fact, nearly all of the decrease in basal melt can be attributed to the loss of sea ice area to melt (following the methods of Bateson et al. (2020)). This suggests that the limiting factor in total melt in the Southern Hemisphere is likely the amount of sea ice, rather than the available heat in the ocean. This is consistent with the larger ocean heat flux convergence ($Q_{flx}$) prescribed in the model for the Southern Ocean compared to the Arctic. Increasing the lateral melt rate results in similar rates of heat flux from the ocean to the ice in most areas of the Antarctic, but over the smaller resulting ice-covered area (Fig. 9). This results in a relatively uniform observed decrease in sea ice concentration across the ice-covered area (Fig. 9b-c). In contrast, in the Arctic, increasing lateral melt 100x increases the rates of heat flux from the ocean to the ice in the summer, but over a smaller area. The limiting factor in melt in

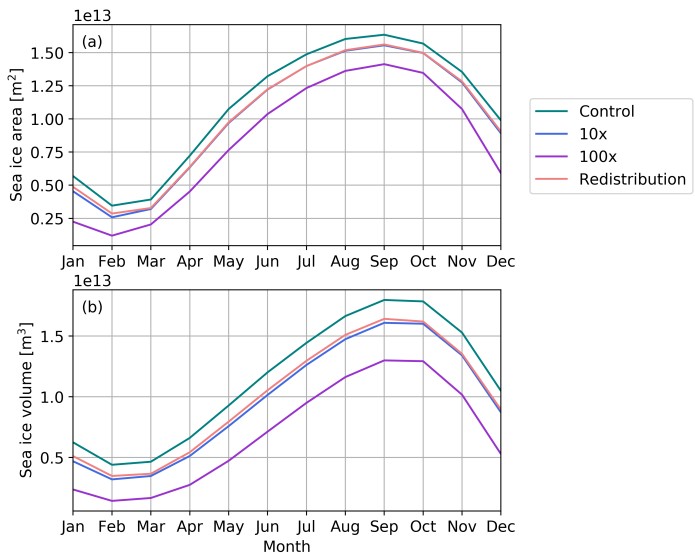

**Figure 8.** Seasonal cycle of Southern Hemisphere (a) sea ice area (m$^2$) and (b) sea ice volume (m$^3$) for all sensitivity runs with pre-Industrial forcing - control run (teal), 10x lateral melt run (blue), 100x lateral melt run (purple), and redistributed lateral melt (light red). Results shown are averages over the last 5 run years for the redistributed lateral melt run, and the last 25 years for all others.

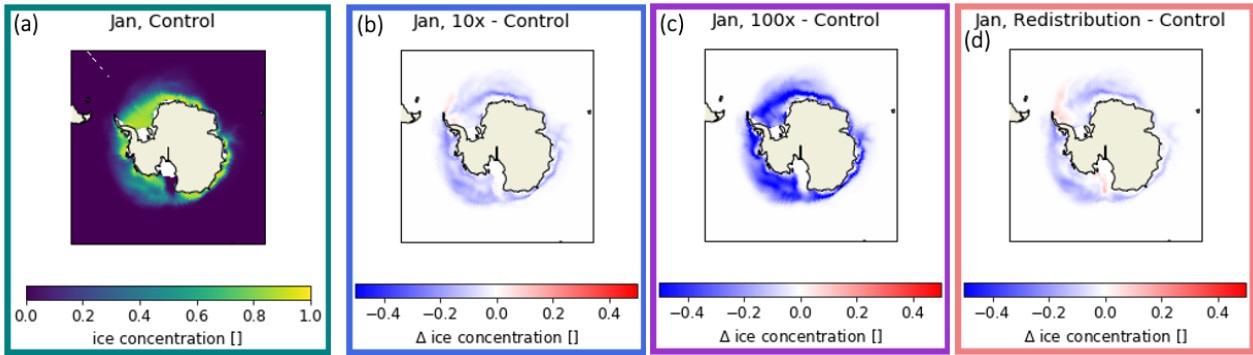

**Figure 9.** Maps of January Southern Hemisphere sea ice concentration (a) in the control run, and relative change in the (b) 10x, (c) 100x, and (d) redistributed lateral melt runs.

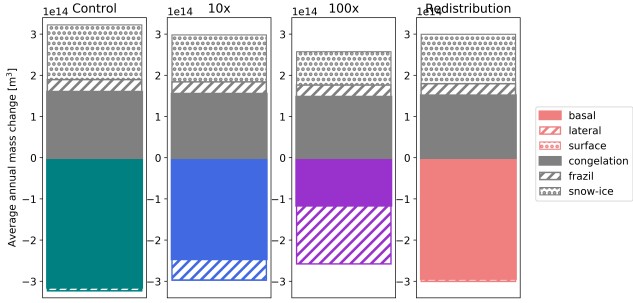

**Figure 10.** Annual Southern Hemisphere mass budgets for all sensitivity runs (from left to right: control, 10x lateral melt, 100x lateral melt, redistributed lateral melt). Grey areas represent volume of growth terms - congelation (solid), frazil (hashed), and snow-ice (dotted) - and colored areas represent volumes of melt terms - basal (solid), lateral (hashed), and surface (dotted). Results shown are averages over the last 5 run years for the redistributed lateral melt run, and the last 25 years for all others.

the Arctic Ocean is the available ocean heat, and the reduction in ice area compensates for the increased ice-ocean heat flux to result in a similar total annual melt.

The OWFE of lateral melting in the Antarctic is higher early in the melt season, when open water likely contributes to larger ice-albedo feedback (Fig. 11). The OWFE of vertical melting for the control run peaks in early January, when it becomes more efficient than lateral melting. During January to February, the thin ice is melting more rapidly and the average thickness of the ice is increasing. The OWFE of lateral melting and vertical melting diverge more strongly in the 100x run (magenta lines in Fig. 11). Here, lateral melting has a higher efficiency at forming open water than vertical melting through nearly the entirety of the melt season. As the average ice thickness peaks in mid-February, and the average ice thickness begins to decline due to the formation of new ice in some regions, the vertical melt OWFE increases again. Similar shapes are seen across the other runs. The increased absorption of shortwave radiation due to an ice-albedo feedback is likely to play a role in delaying the ice growth during freeze-up. The differences observed in the role of lateral melting in the Antarctic compared to the Arctic may be indicative of what we might expect to see for Arctic sea ice in the future, when an ice-free summer is common and sea ice is even more seasonal.

### 3.5 Sensitivity in a 2xCO$_2$ scenario

We expect melt processes to be different in other climate conditions with less overall sea ice, particularly in the Arctic where there is more often multi-year ice. We completed sensitivity runs with control and 100x lateral melt rate settings in a 2xCO$_2$ scenario. Runs were branched from an experiment with atmospheric carbon dioxide abruptly increased to 2x pre-industrial levels that has been run to steady state.

There is lower sea ice area and volume from November to July in a 2xCO$_2$ scenario with 100x the default lateral melt rate (Fig. 12). Both the control and 100x runs experience an "ice-free summer", with effectively no sea ice from July through October (Fig. 12a). Compared to the runs with a pre-industrial forcing, there is less total and percent change from increasing

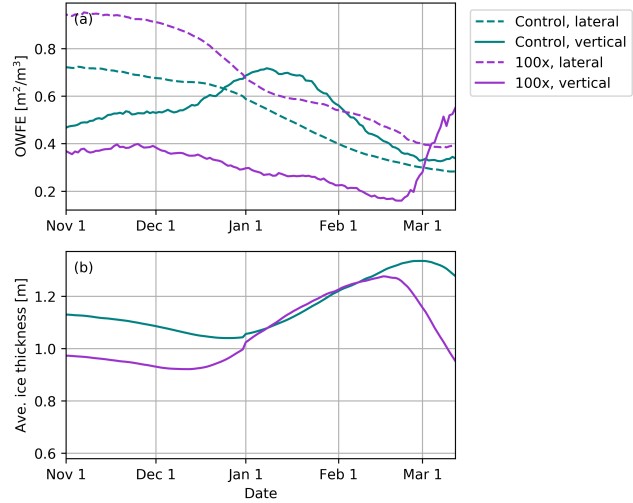

**Figure 11.** Daily (a) open water formation efficiency (OWFE) of lateral and vertical melt and (b) basin-averaged sea ice thickness in the Southern Hemisphere over the melt season. Results are shown for control and 100x lateral melt runs only (teal and purple, respectively), and are averages over the last 25 run years.

the lateral melt 100x as the sea ice is already substantially less extensive and thinner. Maximum sea ice area similarly occurs in March, but is approximately 50% lower than in pre-industrial conditions. Most notably, the increased lateral melt rate steepens the decline to ice-free summer conditions during the melt season (Fig. 12).

Both total and process OWFEs are significantly higher for both runs; while total OWFE was below 0.4 for all pre-industrial runs, it is approximately 0.8 for both in the $2xCO_2$ runs (Fig. 13a). Separating it out into lateral and vertical melt shows that the 100x lateral melt rate significantly increases the OWFE of lateral melt and decreases the OWFE of vertical melt processes. Interestingly, the OWFE is higher for vertical melt than lateral for the control run. The low average ice thickness allows vertical melt processes to be very efficient at forming open water, and so lateral melt is relatively inefficient.

In $2xCO_2$ runs, lateral melt dominates open water formation early in the summer (regardless of the lateral melt rate). OWFE of vertical melt peaks just before the Arctic becomes essentially ice-free, when ice is very thin (Fig. 13b). The difference between lateral and vertical melt during early summer melt is enhanced by increased lateral melt rates. This suggests that in thinner ice scenarios, lateral melt is more critical to open water formation feedbacks driving ice-free Arctic conditions. In particular, as Arctic sea ice becomes thinner on average over the 21st century, increasing the parameterized lateral melt rate

may result in earlier ice-free conditions as a result of an earlier peak in open water formation efficiency.

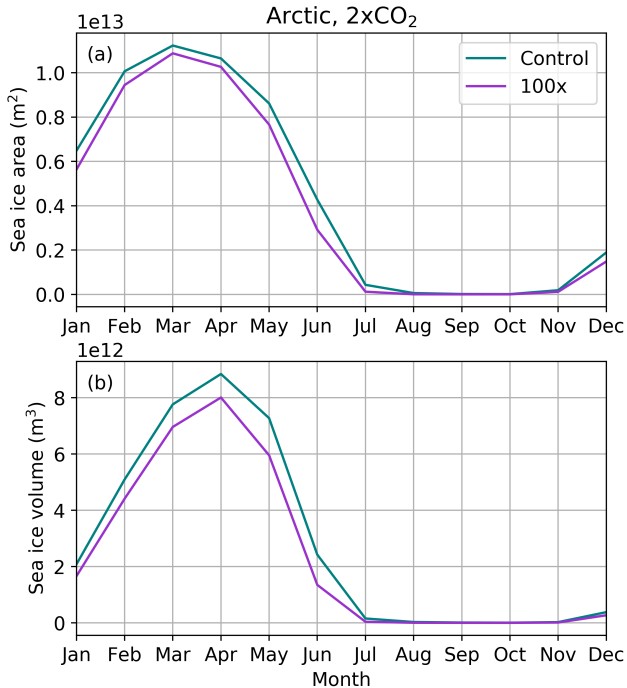

**Figure 12.** Seasonal cycle of Northern Hemisphere (a) sea ice area ($m^2$), (b) sea ice volume ($m^3$) with $2xCO_2$ forcing - control run (teal), and 100x lateral melt run (purple). Results shown are averages over the last 25 run years.

## 4 Discussion

### 4.1 Role of the ITD

Though values of OWFE are not able to be observationally determined, we suggest that one factor contributing to why we don't see a large change in sea ice mean state with initial increases in lateral melt rates in the model (i.e., 10x run) is because of the representation of the ice thickness distribution (ITD). The range of sea ice thicknesses can be represented by a discretized ice thickness distribution (ITD), where growth moves ice into thicker bins, and melt moves it into thinner ones (Thorndike et al., 1975). Most simply, the ITD can be represented by a fixed thickness in each category, but this can result in unrealistically high diffusion of sea ice towards the ends of the distribution. The ITD in CICE assumes a linear distribution of thicknesses within each thickness category and uses a linear remapping scheme to move ice melted (and grown) between categories (Lipscomb, 2001). As a result, melting in the thinnest category generally results in the formation of some open water, even from basal and surface melt. Thus, the model's representation of ice thickness may contribute to vertical melt being relatively efficient at forming open water (and thus having a high OWFE).

The details of the ITD, including the linear remapping scheme and the number and bounds of thickness categories, are likely to have an impact on melt efficiency and thus the ice mean state. Prior studies have investigated the impact of the number of

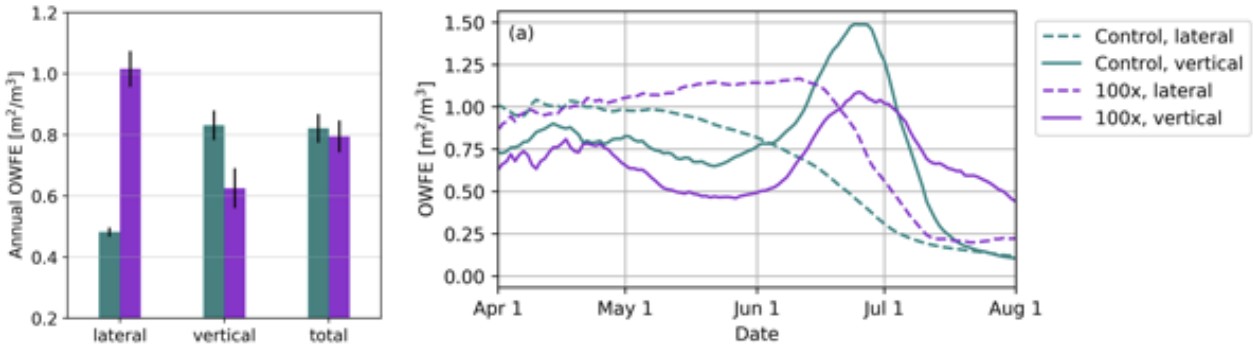

**Figure 13.** Open water formation efficiency (OWFE) in the Northern Hemisphere for $2xCO_2$ forcing runs. (left) Annual average OWFE for lateral melt, vertical melt processes, and total melt. Colored bars show the 25-year mean, with control run in teal, 10x lateral melt run in blue, 100x lateral melt run in purple, and redistributed lateral melt in light red. Black bars denote one standard deviation. (right) Daily OWFE throughout the Arctic melt season of lateral melt (dashed lines) and vertical melt (solid lines) for control and 100x lateral melt runs (teal, and purple, respectively).

ITD categories in climate models with sea ice coupled to the ocean (Ungermann et al., 2017; Massonnet et al., 2011; Moreno-Chamarro et al., in review, 2019; Bitz et al., 2001). Their results generally suggested no clear benefit from increasing the number of categories beyond five, and that representation of thick ice categories has the most impact on the representation of Arctic ice in the current climate. However, we might expect that the resolution of thin ice categories will have more impact in a fully coupled climate model, where ice-albedo feedbacks are likely more realistic as the atmosphere can respond to changes

in the ocean and sea ice. More work to understand the role of the number of ITD categories and remapping scheme in a fully-coupled context is needed. Particularly, it will be informative to investigate how the ITD controls feedbacks related to melt processes.

## 4.2   Implications for future parameterization development

Putting the magnitude of the sensitivity to lateral melting in the context of model sensitivity to other sea ice parameterizations

is instructive in understanding the relative importance of this process. For example, small changes to the sea ice albedo have large impacts on the sea ice mean state. Here we used tuned albedo values that were determined to produce a more realistic historical sea ice state (Kay et al., in review). The small changes to albedo of snow on sea ice (detailed in Sect. 2.1) resulted in an increase in sea ice volume that is quite comparable to the reduction in volume associated with increasing lateral melt rate 100x. In other words, the loss of sea ice volume resulting from 100x increase in lateral melt rate is approximately the same as

that resulting from increased surface melt with a small decrease in snow albedo. Additionally, a similar magnitude of reduction in sea ice volume is achieved by reducing the magnitude of snow on sea ice by half (Holland et al., 2021). The role of lateral melting in achieving a realistic sea ice cover with the appropriate feedbacks will clearly depend on reasonable sea ice albedo and snow cover.

As the results suggest that nuances of parameterizations can impact the sea ice state, we encourage revisiting model representations of how ocean heat drives sea ice melt using observations. Other factors beyond those currently included are likely to contribute. Modeling work (Skyllingstad et al., 2005) has suggested that lead width and wind speed are likely to influence the melt rates. Forthcoming observations from MOSAiC (Nicolaus et al., 2021) and other summer field campaigns will be useful to better constrain the controls on melt processes. For example, Richter-Menge et al. (2001) presented observations from the SHEBA campaign in 1998 that showed the accumulation of significant heat content in the upper meter of a lead. This led to accelerated lateral melt, and rapid basal melt associated with the delayed mixing of this layer. We saw a similar evolution in leads during the summer on the MOSAiC expedition in 2020, with a warmer and fresher layer on the order of 0.1 to 1 m thick spanning more than a month during the melt season. This suggests that small-scale stratification is a common occurrence in summer leads that should be considered in modeling. Currently, the minimum 10 m ocean surface layer in CESM2 prevents simulating the nuances of how much heat is captured specifically in leads. One possible way forward is to explicitly represent aspects of leads within the sea ice component of the model, such that finer-scale processes can be represented.

Another potentially important simplification in the model representation is that it assumes a uniform distribution of vertical and lateral melt locally, as sea ice is represented as a rectangle for thermodynamic and radiative transfer calculations (Fig. 1). In reality, lateral melt rates are often higher near the surface, resulting in a sub-surface shelf. The formation of the shelf, which appears light like a melt pond (e.g., Perovich et al., 2003, Fig. 13), results in a higher albedo locally and likely reduced heat accumulation. In addition, the results suggest that basal and lateral melt rates are closely linked. It could be important to resolve uncertainties in basal melt transfer coefficients, which is beyond the scope of the current sensitivity study. Revisiting lateral melt parameterizations will require finding new ways to capture the important physical controls.

While we do not specifically address here what the correct rate of lateral melting should be, model experiments by Roach et al. (2019) with wave-ice interactions suggest that effective floe diameters as low as 3 m may be realistic through much of the Arctic MIZ, where lateral melt is greatest. Lateral melt rate representation relies on other parameters in addition to floe size, but clearly physically realistic floe size distributions are an important first step in simulating more realistic melt distribution. For example, it will be possible to explicitly represent the relationship of thickness and floe size in CICE using the joint floe size-thickness distribution developed by Roach et al. (2018) fully coupled with wave, atmosphere, and ocean models. There are a number of additional efforts currently underway to include floe size distributions in sea ice models (e.g., Zhang et al., 2015; Roach et al., 2018, 2019; Boutin et al., 2020, 2021; Bateson et al., 2020). Thus, the redistribution of lateral melt towards thinner categories as tested in this study may be able to be explicitly represented in future versions of the model.

# 5 Conclusions

We have assessed the sensitivity of sea ice to the lateral melting parameterization in a coupled climate model. Our sensitivity runs in pre-industrial and $2xCO_2$ climates confirm the importance of a physically based parameterization for lateral melting through a few key conclusions:

- Higher lateral melt rates increase the efficiency of lateral melting at forming open water, which decreases local ice concentrations during the summer. Due to the interconnected nature of lateral and basal melting, this does not uniformly result in less ice. Feedbacks related to open water formation result in a notably thinner and less extensive ice cover in the Arctic (particularly at high lateral melt rates) and in the Antarctic.

- The details of how lateral melting is represented and distributed matter. The assumption that lateral melting occurs at the same rate across all ice thicknesses is particularly called into question, as it is unlikely to capture the complexity of feedbacks associated with melt. Higher lateral melt rates (or smaller floes) in areas of thin ice reduce sea ice volume in both hemispheres. Observational constraints on the lateral melt rates are particularly necessary to constrain the possible impact of increased lateral melt rate on thinning ice throughout the 21st century.

- While the lateral melt may play a relatively small role in the mass budget, these results suggest that this does not tell the full story of how sea ice evolves thermodynamically. Lateral melt can have an impact on the mean state due to its role in open water formation which is key to the ice-albedo feedback.

The applicability of the results presented here may be limited by the sophistication of the model used. The use of the slab ocean model may result in the inability to represent some complexities in feedbacks due to the lack of seasonally and inter-annually varying MLD, and prognostic dynamic ocean heat transport. As a result, we may miss additional feedbacks associated with thermodynamic changes, such as shifts in brine rejection during freeze-up, and changes in ocean dynamics. In the future, a fully coupled dynamic ocean model could be run to understand and isolate the role of such factors. Additionally, there is a great deal of uncertainty in the current functional form of the lateral melt parameterization. Some of the suggested conclusions may require revision due to the future incorporation of a floe size model, which will more accurately capture the complexity of feedbacks with changes in open water formation.

*Author contributions.* MH and BL developed the concept and obtained funding for this work. MS and MH designed and ran experiments. MS led preparation of the manuscript, with feedback from MH and BL.

*Competing interests.* The authors declare that no competing interests are present.

*Acknowledgements.* This research was supported by NSF grants 1724467 and 172474. The CESM project is supported primarily by the National Science Foundation. Computing and data storage resources, including the Cheyenne supercomputer (https:// doi.org/10.5065/ D6RX99HX), were provided by the Computational and Information Systems Laboratory (CISL) at NCAR. We thank all the scientists, software engineers, and administrators who contributed to the development of CESM2. We wish to acknowledge Dave Bailey for assistance with setting up runs and processing outputs. We would like to thank the two reviewers and editor whose feedback substantially improved the quality of the manuscript.

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
