# Peer review of "Arctic sea ice sensitivity to lateral melting representation in a coupled climate model"

_The Cryosphere, 2021_

## Author Comment (AC1)

*We gratefully acknowledge both reviewers' time and useful suggestions. General and point-by-point responses to comments describing changes made in the revised manuscript are provided below in italics.*

Anonymous Referee #1

**General Comments**

This paper revisits the longstanding parameterisation of lateral melt used within the CICE (Los Alamos) sea ice model and explores how assumptions about the representation of lateral melting impacts sea ice within a coupled climate model (in this case CESM2.0). As the authors note, significant progress has been made recently in terms of representation of floe size within sea ice models, but there has been a corresponding lack of attention paid to the lateral melting parameterisation. Whilst other studies have explored similar themes exploring sea ice model sensitivity to floe size and lateral melting e.g. Steele et al. (1992), and more recently Bateson et al. (2020), this is the first study I am aware of that addresses the assumption of a constant lateral melt rate across all sea ice thickness categories. The application of the concept of open water formation efficiency to provide further understanding of how lateral and basal melting processes impact the sea ice cover is a particularly strong and valuable feature of this work. I therefore believe that this paper initiates an important and valuable discussion into the lateral melting parameterisation in sea ice models and will make a valuable contribution to the literature.

The scientific quality of the work presented is generally strong, with good associated analysis and discussion. The methodology could be more thorough in terms of the details provided, and there is perhaps insufficient consideration of the limitations of the conclusions reached. I am also unconvinced that some aspects of the conclusions reached are justified by the results presented and these either need to modified or further evidence provided. The figures are of a good quality and appropriate to the discussion. I have suggested a couple of additional figures that might be helpful to illustrate some of the discussion, but this is not essential. Similarly, the structure generally seems fine, though I do have some questions about whether some of the discussion should be moved to the results section, and some of the conclusion section then moved to the discussion. The paper reads well, is clear in its conclusions, and also has a representative abstract and title.

Overall, I believe that this paper is within the scope of The Cryosphere and, with some moderate edits, merits publishing.

*The "Model and experimental design" section has been expanded in response to the reviewer's comments, which are described in more detail in the point-by-point responses below. The use of the SOM allows us to do multiple coupled climate runs that would be cost-prohibitive with a fully coupled ocean - this point has been expanded on in 2.1. "The use of the SOM requires significantly less computational time both because it allows the model to converge must faster (e.g. around 30 years vs. 100s of years for the fully-coupled model), and as a result of not running a full ocean model. Thus, implementing the SOM allows us to run multiple sensitivity tests in a coupled climate that would be prohibitive with the fully-coupled dynamic ocean." We have also added a paragraph to the conclusions addressing some caveats, and taken the reviewer's suggestion to add additional figures showing the spatial distribution of changes in concentration. The material in the results, discussion, and conclusion has been re-organized based on the reviewer's comments.*

| | Specific Comments | Response |
|---|---|---|
| 1-1 | P2 L52-53: Could you add further details on what you mean by the following: 'such as related to model resolution,'? | *This sentence has been removed. The compensating effect we wished to introduce here is summarized in the next paragraph: "[Bateson et al., 2020] found that increased lateral melting was largely compensated for by decreased basal melting in standalone sea ice models."* |
| 1-2 | P2 L58: Re following, 'i.e. Bateson et al., 2020)'. e.g. would probably be better here rather than i.e. since the study referred to is one of several on this theme. | *This reference has been removed from this sentence, and a more complete list has been added earlier in the paragraph (e.g. Zhang et al., 2015; Roach et al., 2018, 2019; Boutin et al., 2020, 2021; Bateson et al., 2020).* |
| 1-3 | P3 L62-64: Re following statement, 'different results might be expected in a coupled climate model that allows feedbacks related to the formation of open water'. There is some evidence of this in Fig. 5 in Roach et al. (2019). Simulations with a standalone sea ice model generally showed a reduction in lateral melt and increase in basal melt of comparable magnitude, but in a coupled sea ice-ocean setup the reduction in basal melt was significantly smaller than the increase in lateral melt. Might be worth referring to this? | *Thank you for catching this. We have edited this to the following: "They found that increased lateral melting was largely compensated for by decreased basal melting in standalone sea ice models, but the reduction in basal melt was smaller than the increase in lateral melt in a coupled sea ice-ocean setup (Roach et al., 2019). Even further differences might be expected in a model with a coupled atmosphere that allows feedbacks related to the formation of open water."* |
| 1-4 | P3 L74-76: I am unconvinced you have achieved the final aspect of this objective with the results presented: 'as a result of ice-albedo feedback'. Later comments will further address this. You may need to modify this paragraph depending on how you decide to address some of the later comments. | *We agree with the reviewers comment that we are not able to show that the ice-albedo feedback is the primary driver for observed changes, and we are also interested in understanding the impact related to other changes such as the shift in mean state. As a result we have softened the language here to "…as a result of factors driving sea ice change associated with open water formation, including the ice-albedo feedback", and elsewhere where relevant.* |
| 1-5 | P3-4 L79-96: I think in general this section would benefit from a more complete discussion of details of the model setup that are pertinent to this study e.g. additional details of the SOM (given the importance of surface ocean properties to lateral and basal melt rates), a more complete description of the forcing and how it is applied, and details on how the sea ice is initiated. | *We have added to this section: a more complete description of the SOM, including the defining equation (Eq. 1); and a brief description of preindustrial control forcing and branching* |
| 1-6 | P3 L80-89: Given the significant focus in this paper on the ice-albedo feedback, I think some | *In addition to softening the focus on the ice-albedo feedback throughout the paper (see* |

| | | |
|---|---|---|
| | discussion is required here or elsewhere about the possible impact of using a prescribed mixed-layer depth without a full representation of sea ice-ocean feedbacks. | *response to 1-4) we have edited the methods to more clearly describe the implications of using a slab ocean, including, of particular relevance for this comment: "The mixed layer temperature (SST) evolves with surface heat fluxes determined by the coupled climate model; thus ice-albedo feedbacks are permitted."* *Additional discussion of the caveats resulting from the use of SOM has been added to the conclusions.* |
| 1-7 | P3 L85: Re following statement, 'although not specifically constrained in the model'. Can you clarify what you mean by this? | *This sentence has been removed and replaced by "The primary limitation with the SOM is that there is no ability for ocean dynamics variability to drive changes in ocean heat content." This sentence better highlights the most important difference and limitation of the SOM.* |
| 1-8 | P4 L93-94: It would be helpful to add a brief comment on the tuned albedos. How are they different from standard values used? | *The following sentence was added: "Specifically, the albedo of snow on sea ice was increased by decreasing the snow grain radius (with an increase of the parameter r_snw from 1.25 to 1.5), and the temperature at which we allow snow grain growth to occur (due to melting conditions) was increased by 0.5 degrees C from 1.0 C to 1.50 C."* |
| 1-9 | P5 L127: Re following statement, 'if it does, reductions are made to the lateral and basal melt rates by a constant factor'. A more detailed explanation would be helpful here on how the limits to the lateral and basal melt rates are calculated and applied. | *The sentence has been expanded to "If the sum of lateral and basal heat flux represents a larger flux than that available, the lateral and basal melt heat fluxes are both reduced by the scalar factor necessary such that all ocean heat content is lost to the ice."* |
| 1-10 | P6 L144: Can you provide more details on why you specifically selected this form of lateral melt redistribution (as opposed to an inverted $r_n$, or higher / lower spread of values for $r_n$)? | *See response to 2-5. Additionally, we now clarify that: "The focus here was on making simple changes to understand the impact of the limitations in the lateral melting parameterization itself, rather than prescribing what an appropriate distribution of lateral melting should be."* |
| 1-11 | P6 L145: Re following statement, 'these values were distributed around 1 with the aim of keeping the total lateral melt volume approximately the same, such that the effect of the redistribution can be uniquely observed'. Does this not rely on an equal distribution of ice volume between thickness categories? In locations dominated by thin or thick ice, would | *Indeed, we do not expect this run to ultimately retain the same lateral melt due to unequal distribution across ice thickness categories (as noted by the reviewer) and resulting changes in the mean ice state. In fact, we do see the largest changes in areas of thinnest ice (Fig.* |

| | | |
|---|---|---|
| | this setup not produce abnormally high or low lateral melt rates? | *4d) but this is tied to the physical justification for redistribution scheme (see response to 2-5).*

*We considered a redistribution scheme that was weighted based on the initial ice thickness distribution, but the variability across seasons and regions did not necessarily result in a more physically meaningful parameterization than the one chosen. Overall, we acknowledge that this is not necessarily the best or only way to test this impact, but is one way of implementing a simple change that demonstrates the limitations of the parameterizations.* |
| 1-12 | P7 L179-180: Re following statement, 'lateral melting rate is applied to all categories equally'. You should clarify that this is for the standard lateral melt parameterisation only, not the simulation using Eq. (4). | *This has been clarified by changing the referenced statement to "In the standard lateral melting parameterization, the lateral melting rate is applied to all categories equally. While sea ice melts equally in all categories, thin ice categories will form open water most rapidly such that the OWFE of lateral melt should be directly tied to the average thickness."* |
| 1-13 | P8 L196-197: Re following statement, 'contrary to intuition, increasing the lateral melt does not necessarily reduce sea ice area and volume'. My understanding from Fig. 3 is that in both simulations where the lateral melt rate is increased, the sea ice area is reduced, and the same is true for volume from March to August? I think this statement should be reworded to better reflect the results presented in Fig. 3. | *Thank you for this point. The sentence has been edited to "Contrary to intuition, substantial increases in the lateral melt rate do not necessarily result in reductions of sea ice area and volume of a similar magnitude."* |
| 1-14 | P9 L202-205: A map plot showing differences in sea ice concentration might be useful here to illustrate how the differences vary across the sea ice cover. | *Maps of change in early summer sea ice concentration (relative to the control run) in both hemispheres are now included in the manuscript.* |
| 1-15 | P9 L211-212: Have you done any analysis of the model output to confirm that the available heat content in the surface ocean is the limiting factor for basal / lateral melting? This is not the only mechanism for the basal melt compensation effect in response to an increase in lateral melt in sea ice models e.g. in Bateson et al. (2020), it is demonstrated that the primary mechanism in standalone CICE is from the physical reduction in available sea ice area for basal melt (see Figs 4-5 in that paper). I | *We would like to thank the reviewer for this helpful suggestion. We have now completed the analysis to test this and found that the physical reduction in available sea ice constitutes a significant part of the compensation effect in our coupled runs.*

*The text has been edited to acknowledge this additional mechanism: "The change in the sea ice area also contributes to the observed basal melt decrease. Following the analysis by* |

| | | |
|---|---|---|
| | think you either need to do some additional analysis to confirm that the mechanism suggested is the primary mechanism driving the basal melt compensation effect or acknowledge that it is not the only possible mechanism. | *Bateson et al. (2020, Fig. 5), we find a partial contribution from loss of ice area (for example, in July it accounts for 33% for 10x run; 48% for 100x run) compared to the nearly complete attribution in their standalone sea ice model runs." Maps of the predicted (synthetic) and modeled (actual) reduction in basal melt are shown below for reference, but not included in the text.*
[Figure]
 |
| 1-16 | P10 L221-222: I do not think you have presented sufficient evidence to demonstrate a significant ice-albedo feedback effect. There are other mechanisms that could plausibly result in a change to the sea ice mean state, particularly for simulations evaluated over decadal timescales e.g. a change in how sea ice is distributed between thickness categories, particularly since sea ice vertical growth rates are sensitive to the existing sea ice thickness, or more efficient use of available surface ocean heat content for sea ice melting. Can you directly isolate and quantify the change in sea ice mean state that can be attributed to the ice-albedo feedback mechanism here? Otherwise, you should acknowledge that the ice-albedo feedback is not the only possible mechanism that could cause a change in the mean sea ice state, and further analysis / studies are required to quantify its impact. | *See response to 1-4 for a more general response to this comment. A formal feedback analysis is not appropriate here, so we have changed the text at the referenced point to: "As the lateral melt rate becomes much larger, there is more open water formed in ice covered areas over the melt season (Fig. 3 c) and the feedbacks are significant enough to result in substantial changes in sea ice mean state. This likely includes the ice-albedo feedback, as well as other processes related to dynamic and thermodynamic changes in the ice."* |
| 1-17 | P10 L223-227: Or due to non-equal distribution of sea ice volume between thickness categories? See earlier comment. | *The referenced text has been edited to: "This is likely a result of the adjustments to the lateral melt parameterization (Fig. 2) and the unequal* |

| | | *distribution of sea ice between categories; the redistribution results in more rapid melt of thin ice, which results in less thin ice to melt, and there is less lateral melting in the thick ice that remains."* |
|---|---|---|
| 1-18 | P13 L298-300: Re following statement, 'increasing the lateral melt rate results in similar rates of heat flux from the ocean to the ice in most areas of the Antarctic, but over the smaller resulting ice-covered area (not shown)'. A map plot would be useful here to illustrate this point. | *All data needed to perform heat budget analysis and show changes in heat flux are not available from these simulations, and computing limitations prevent us from re-running to include. We have now included maps of sea ice concentration which show that there is a smaller resulting sea ice-covered area, and that there is "a relatively uniform observed decrease in sea ice concentration across the ice-covered area (Fig. 9 b-c).* |
| 1-19 | P13-14 L310-311: Re following statement, 'here, ice-albedo feedback is not the main reason for why increasing lateral melting results in lower sea ice mean state.' I may have missed or misunderstood something here, but it is not clear to me what you propose as the mechanism driving changes in the Antarctic sea ice mean state. | *Following on the response to comment 1-15, we have also added here the analysis from Bateson, which shows that: "In fact, nearly all of the decrease in basal melt can be attributed to the loss of sea ice area to melt (following the methods of Bateson et al., 2020). This suggests that the limiting factor in total melt in the Southern Hemisphere is likely the amount of sea ice, rather than the available heat in the ocean." Additionally, the sentence referenced in the comment has been removed from the text.* |
| 1-20 | P17 L352-355 & L359-361: As discussed above, I think you need to modify these conclusions given there are plausible mechanisms other than the ice-albedo feedback to explain why increases in lateral melt change the mean sea ice state. | *Language in 1st conclusion has been edited: "…ice albedo feedback results in…" has been changed to "feedbacks related to open water formation result in…"*
*and the 3rd conclusion has been softened to: "…Lateral melt can have an impact on the mean state due to its role in open water formation **which is key to** the ice-albedo feedback"* |
| 1-21 | P18 L387-388: I suggest you put e.g. in the list of references here, given this is a non-exhaustive list of the different FSD model developments in existence. | *Done. The reference list has also been expanded to include a more complete representation of the current literature on this topic.* |
| 1-22 | General comment about paper structure: It is not obvious to me why section 4.1 and 4.2 (particularly the former) are classified as discussion sections rather than results sections. Similarly, the final three paragraphs in the conclusions section could be moved to the | *The reviewer makes a good point. Subsections "Sensitivity in the Southern Hemisphere" and "Sensitivity in a 2xCO2 scenario" have been moved to the Results. The discussion from what was formerly the conclusion has been moved to a new sub-section in the Discussion* |

| | | |
|---|---|---|
| | discussion section since they introduce new material and discussion. | *"Implications for future parameterization development".* |
| 1-23 | General comment about conclusions: It would be useful to have some reflection on the limits of these conclusions e.g. the limitations of using the SOM. | *We have added a paragraph to the conclusions discussing the caveats associated with limitations of the SOM.* |
| | **Technical Corrections** | Response |
| 1-24 | P1 L11: The phrase 'well representing' here is somewhat awkward. Maybe replace well with accurately? | *Done* |
| 1-25 | P2 L36: Should be 1980s, rather than 1980's. | *Done* |
| 1-26 | P2 L38: The )'s setup of 'Josberger and Martin (1981)'s formulation' is awkward. Maybe replace with 'the formulation of Josberger and Martin (1981)'. | *Done* |
| 1-27 | Figure 2 caption: 'ncat' is not referred to or defined anywhere else in this manuscript. | *This term has been removed from the caption.* |
| 1-28 | P4 L108: Maybe replace 'Lipscomb (2001) (Eq. 22)' with 'Eq. (22) in Lipscomb (2001)'. | *Done* |
| 1-29 | P5 L124: I do not think you have defined $V_{ice,n}$ in this equation. | *Thank you for catching this. A definition has been added immediately following the equation (Eqn. 3).* |
| 1-30 | P6 L135: In some places you have not followed The Cryosphere journal style guide e.g. here Eq. 3 should be Eq. (3), and section 2.3 below (L148) should be Sect. 2.3. Also, Fig 2 should be Fig. 2 on L150, and Figure 1 should be Fig. 1 on P7 L179. Similar issues are present elsewhere.. | *All instances of section references have been corrected to "Sect."; all instances of figure references in text have been corrected to "Fig." unless at the beginning of a sentence ("Figure"); all equation references have been corrected to the format of "Eq. (#)" or "Eqs. (#) and (#)". Have similarly confirmed adherence to other style guidelines* |
| 1-31 | P6 L147: Should this be 'per unit volume' rather than 'per volume'? | *Yes, this has been corrected.* |
| 1-32 | Figure 3 caption (and other figures): it would be helpful to clarify the number of years the results have been averaged over in the figure caption. | *"Results shown are averages over the last 5 run years for the redistributed lateral melt run, and the last 25 years for all others." was added to the caption for Figures 3, 4, 5, 7, and 8, and "Results shown are averages over the last 25 run years." was added to the caption for Figures 6, 9, and 10.* |
| 1-33 | P10 L230: Should 'open water efficiency' be 'open water formation efficiency'. | *Corrected* |
| 1-34 | P11 L280: Seasonal should be season? | *Done* |

Anonymous Referee #2

This paper assesses the importance of lateral melting in the context the coupled model CESM. By modifying the parameterization of lateral melting in a coupled model the authors can quantify the contribution of this process in the albedo feedback mechanism. The authors present an elegant quantification of the relative impact of lateral melt in term of efficiency at forming open water compared to the bottom melt efficiency.  This is potentially interesting for climate applications via the well known albedo feedback process. I would like the authors to discuss further this impact in terms of a potential increased contribution of lateral melt throughout the 21st century. I am also a little worried that the sensitivity study proposed is not well justified (why is it ok to vary the lateral melt scale by a factor 100) and not very well constrained by observations (in situ constraints or satellite observations of FSD). I am a little worried as well that the results presented here are only representative of this specific model configuration and in particular of the prescribed mixed layer depth (this could be particularly problematic for the Southern Ocean). As is often the case with such modelling sensitivity studies the paper asks more questions than offers insights and answers. The paper introduces a lot of model experiments but they remain at a qualitative level (arbitrary r_n function, extreme sensitivity lateral melt scale factor, unclear partition between lateral and bottom melt, prescribed mixed layer). As such it could be argued that the paper is more suitable for a modelling journal such as Ocean Modelling or GMD. In short I find the authors have presented an elegant modelling study of the contribution of lateral melt to the open water formation efficiency, are clearly well versed in the workings of the model (CICE/CESM), and in the processes controlling lateral melting but do not offer a significant new model development or model constraint from observations. Provided the authors improve on some (most) of the general comments below the paper could make a useful contribution to the community. Alternatively the authors could resubmit this fine modelling study to a modelling journal.

*In response to the suggestion to discuss the impact of a potential increase in lateral melt contribution in the $21^{st}$ century, we expanded the discussion of the $2xCO_2$ runs to include: "In particular, as Arctic sea ice becomes thinner on average over the 21st century, increasing the parameterized lateral melt rate may result in earlier predicted ice-free conditions annually as a result of an earlier peak in open water formation efficiency." We have also added to the conclusions: "Observational constraints on the lateral melt rates are particularly necessary to constrain the possible impact of increased lateral melt rate on thinning ice throughout the 21st century."*

*We disagree that the paper is more suitable for a modeling journal. As the reviewers comments, the manuscript "do[es] not offer a significant new model development or model constraint"; rather, our aim is to understand how lateral and basal melting processes impact the ice cover through the role in open water formation, and particularly the implications associated with a uniformly applied lateral melt rate. To our knowledge, this is how lateral melting is parameterized in all climate models that currently represent lateral melting (Keen et al., 2021; see Sect. 2.1.2). Thus, the sensitivity studies completed are well justified for this purpose. More discussion on this is provided in the response to individual comments below.*

| | General comments | Response |
|---|---|---|
| 2-1 | Highlight main results better in the abstract (rewrite) as it is too generic at the moment | *The abstract has been edited to more specifically summarize the presented results,* |

| | | including the addition of the following sentences:

*"The more seasonal Southern Hemisphere ice cover undergoes larger relative reductions in sea ice concentration and thickness for the same relative increase in lateral melt rate, likely due to the hemispheric differences in the role of the sea ice-upper ocean coupling. Additionally, increasing the lateral melt rate under a 2xCO$_2$ forcing, where sea ice is thinner, results in a smaller relative change in sea ice mean state, but suggests that open water formation feedbacks are likely to steepen the decline to ice-free summer conditions."* |
|---|---|---|
| 2-2 | Include in the introduction a review of how lateral melting vs vertical melting is currently represented in CMIP6 (Keen et al 2020) | *The following has been added to the introduction: "Of particular relevance here, seven of the 15 CMIP6 models reviewed by Keen et al. (2021) had no explicit representation of lateral melt. There is wide model variation in the partitioning of mass flux between melt processes, but the multi-model mean allocates only 4% to lateral melt while the vertical melt processes account for a combined 77%, with great spread in the relative ratio of surface to basal melt (Keen et al., 2021)."* |
| 2-3 | You use factors 10x and 100x without much discussion as to the validity of such choices. This correspond to changing the mean floe size by a factor 10 or 100 which is consistent with spatial gradients from pack ice to MIZ. Please explain all this a bit more and why a spatially constant scaling makes sense in your view. Discuss also impact of FSD as in Tsamados, Bateson or Horvat. | *The aim is to understand the implications of changes to the currently constant lateral melt rate. To make this clearer, we have added the following to the text: "Here, changes to the floe size are rather used as a catchall for factors impacting the sensitivity. Other parameters controlling the rate of lateral melting as a function of temperature difference, m$_1$ and m$_2$, were kept the same, but it is noted that the effect of increasing m$_1$ is the same as decreasing the diameter D (Fig. 2) such that the 10x and 100x sensitivity runs can alternatively be thought of as m$_1$ = 1.6 x 10$^{-5}$ and m$_1$ = 1.6 x 10$^{-4}$, respectively. There is substantial uncertainty in the default m$_1$ and m$_2$ parameter values, which were derived from a single set of observations (Perovich, 1983). As such, these large perturbations are justified by the uncertainties in the functional form of this parameterization and allow us to look at* |

| | | |
|---|---|---|
| | | *the processes at work and how they impact the sea ice and climate system."* |
| | | *In response to the second part of this comment, we note that we summarize the relevance of this work for the floe size distribution models which are under development (and are still being tested in fully-coupled framework) in the discussion. See also the response to 2-8 for more discussion on this point.* |
| 2-4 | The participation between lateral and bottom melt in CICE (CESM) is not critically reviewed in my opinion | *The following was added to Sect. 2.1.2 Parameterization of lateral melting: "For a selected part of the historical period (1960-1989), CESM2 predicted 0.4 Gt\*10^3 mass loss per year associated with lateral melting, which is about 4.2-4.8% of the total mass loss per year associated with vertical melting (with the range representing configurations with different atmospheric models; Keen et al., 2021) With the pre-industrial forcing and configuration of CESM2 used in this study, the average annual volume loss from lateral melting is similarly about 4.5% of the volume loss associated with vertical melting (Fig. 4)."* |
| 2-5 | The authors introduce an 'arbitrarily' category dependent lateral melt redistribution function r_n. This is qualitative and not robustly justified or quantified. | *Although the specific values used were arbitrarily chosen, due to the lack of observational constraints, the general shape of the redistribution function is physically justified by the three reasons listed in Sect. 2.2, where we hypothesize that lateral melting may be enhanced for thinner ice. The text has been edited to make this justification more explicit.* |
| 2-6 | On a related point this lateral melting scale is a dynamical quantity as more lateral melt leads to a reduction of floe sizes which in turn leads to more lateral melt. I am not sure that you fixed scale approach captures all this complexity and positive feedback. | *We acknowledge that current parameterizations do not capture all the complexity of possible feedbacks, and the examined redistribution has similar limitations. These tests are intended to explore the implications of how lateral melting is parameterized.* *As such, we have added to the abstract: "The runs explore the implications of how lateral melting is parameterized, and structural changes in how it is applied." Additionally, the conclusions state: The assumption that lateral melting occurs at the same rate across all ice thicknesses is particularly called into question,* ***as it is unlikely to capture the complexity of*** |

| | | |
|---|---|---|
| | | *feedbacks associated with melt.*" (bolded text is new).

 *As discussed in Section 4.2, this complexity is more likely to be captured by coupled models with prognostic floe size distribution, which is expected in future model releases.* |
| 2-7 | Explain role of ocean and mixed layer heat reservoir in redistributing between vertical and lateral melt. I would like to see how sensitive your results are to this. With this in mind, are the results for the SO really meaningful (there the MLD can vary a lot and reach 100s of metres - definitely not a constant 10m as in your model) | *The second part of this question indicates that the description of the SOM was unclear and that the reviewer was under the impression that the MLD was defined as 10 m everywhere. This is not the case and has been clarified in the text: "The prescribed MLD varies spatially based on climatological conditions simulated by CESM coupled simulations with an active ocean component. However, it is constant over time (e.g. doesn't vary seasonally or inter-annually) and has a minimum depth of 10 m". This and other changes are intended to clarify that the prescribed ocean mixed layer, with varying temperature, provides the heat reservoir for both basal and vertical melt. Additionally, we have added a paragraph addressing the possible limitations of this sensitivity study associated with the use of a slab ocean to the conclusions.* |
| 2-8 | Can you please constrain your results by comparing to more recent observations of floe size distributions. For example is it possible to assess which of 100x or control is more realistic in terms of relative distribution between vertical and lateral melt? | *Observations of floe sizes are not sufficient to provide constraints on realistic parameterization changes, nor are there observations to determine distribution between vertical and lateral melt. The sparsity of lateral melt observations is discussed in the introduction, and we have added to section 2.2: "Observations of floe size distribution are limited, and do not have sufficient spatial or temporal coverage to determine what floe size is most representative for ice-covered regions; the most complete coverage is provided by satellite products such as CryoSat-2 (Horvat et al., 2019), but the relevance for lateral melt is significantly limited by the footprint of 300 m." Instead, we suggest that the sensitivity tests completed utilize defensible values as "...model experiments by Roach et al. (2019) with wave-ice interactions suggest that effective floe diameters as low as 3 m may be realistic through much of the Arctic MIZ, where lateral melt is greatest." (Section 4.2)* |
| | Specific comments | Response |

| | | |
|---|---|---|
| 2-9 | P1 L24 "Vertical melt processes (surface and basal) can only form open water once the ice is very thin, while lateral melt can directly form open water area regardless of ice thickness" are you aware of MOSAiC experiments planning to re-evaluate the relevance of this statement. | *During MOSAiC we made measurements of surface and basal melt across different ice thickness categories and collected data in order to estimate lateral melt at a number of locations. I do not expect any of this data suggests a different process than has been stated here. Of course, the rate of lateral melting (and thus its ability to form open water) tends to be relatively low, which I expect is what the reviewer is hinting at. This is mentioned later in the introduction, and will hopefully be evaluated more quantitatively with observations from MOSAiC, as suggested in Sect. 4.2 in the discussion: "Forthcoming observations from MOSAiC … will be useful to better constrain the controls on melt processes". In the conclusion, we reiterate the need for "observational constraints on lateral melt".* |
| 2-10 | P3 L81 How critical is the depth of the this SOB for your results? Sensitivity? I.e. how much of the heat in the SOB would have been lost to the lower ocean? | *We again clarify that the SOM does not have a fixed depth of 10 m (see response to 2-7). As we do not have a fully-coupled dynamic ocean run of the same experiment to compare to, we are unable to specifically address the sensitivity to the SOM here. However, similar studies have examined the sensitivity of climate and sea ice to this model setup. For example, we have now added to the text:*
*"Roach et al. (2019) used the SOM for experiments focused on sea ice floe size, justified by qualitatively similar results between the slab ocean model and a fully-coupled dynamic ocean model."*

*Additionally, new text in 2.1 clarifies that the $Q_{flx}$ term in the SOM equation contains all dynamical ocean heat flux terms, and is computed from fully coupled simulations as a residual using the SOM equation. This does account (in a fixed climatological fixed way) for heat being lost from or gained by the fixed-depth mixed layer represented by the SOM.* |
| 2-11 | P4 L104 where all are | *Corrected* |
| 2-12 | P4 L110 cite Massonnet, François, et al. "On the discretization of the ice thickness distribution in the NEMO3. 6-LIM3 global ocean–sea ice model." Geoscientific Model Development 12.8 (2019): 3745-3758. | *Done* |

| | | |
|---|---|---|
| 2-13 | P5 L120 D=300 as a default. Please discuss this approximation and why it could not be turned into a dynamical variables. | *As mentioned in the introduction, it is only in recent model developments that there is the possibility to include prognostic floe sizes. The following sentence has been added to Sect. 2.1.2: "We note again here that while there are recent model developments to include an evolving floe size distribution based on coupled processes (e.g. Zhang et al., 2015; Roach et al., 2018, 2019; Boutin et al., 2020, 2021; Bateson et al., 2020), most models do not currently have the capability to have a variable floe size or include the necessary couplings (such as surface waves)."* |
| 2-14 | P5 L126 in addition | *Done* |
| 2-15 | Figure 2 2 orders of magnitude of changes in the lateral melt rates 'scale' seems very unconstrained by observations to me | *See response to 2-8* |
| 2-16 | P7 L164 to clarify definition of dV/dt_lat,n add these terms in eq (4) | *Eq. (4) (now Eq. (5)) has been edited to include (dV/dt)_lat,n* |
| 2-17 | P7 L165 remove as n already defined | *Done* |
| 2-18 | P7 L169 to the average…in the control | *Corrected* |
| 2-19 | Figure 1 I quite like the drawing but it does not represent 'key melt processes' but rather fluxes and variables of interest. Also I feel that it does not express all the quantities described in the paper as discussed in P7 L179 | *This is a fair point; unfortunately, I am not sure how to truly illustrate melt processes with a drawing! The Fig. 1 caption has been revised to "Schematic of key components of the CICE sea ice model, including the five-category ice thickness distribution, and important fluxes and melt terms." At the referenced text at L179, "…Fig. 1 illustrates…" was changed to "…Fig. 1 helps to demonstrate…"* |
| 2-20 | P7 L180 'open water forms equally in all categories' seems in contradiction with r_n -> clarify this entire last paragraph. | *A number of small changes have been made to clarify this paragraph, mostly notably in response to this comment: "**In the standard lateral melting parameterization**, the lateral melting rate is applied to all categories equally."* |
| 2-21 | P9 L201 minima | *Done* |
| 2-22 | P9 L209 this communicating vase issue is crucial and I am worried that there is not enough discussion on how the relative basal to lateral ratio of melt is affected by the SOB characteristics (depth value) and lack of dynamics (fixed depth) | *I believe this comment stems from lack of clarity on the formulation of the SOM. Please see responses to 2-7 and 2-10 for changes that have been made to address this.* |

| 2-23 | P11 L250 what about then a 10x & distribution sensitivity run | *I believe the reviewer is suggesting a sensitivity run with 10x greater lateral that is re-distributed primarily towards thinner ice categories (as in the redistribution run). As the primary purpose of this study is to assess the impact of the lateral melt parameterization on the coupled climate system, we do not believe such combinatorial sensitivity tests would add additional value.* |
|---|---|---|
| 2-24 | P11 L262 you mean efficiency in terms of open water formation but other aspects might still be affected (more winter growth…) -> clarify sentence | *Indeed, the decrease in thickness could increase the efficiency of winter growth, or other processes. This sentence has been clarified by changing "efficiency" to, more specifically, "open water formation efficiency".* |
| 2-25 | P12 L276 so which of 100x or control is more realistic? Why not do a 1000x run or a 0.1x run as at present you do not seem to constrain these sensitivity runs at all from observations. | *A 0.1x run is not deemed necessary, as the lateral melt is already such a small fraction of total melt in control runs (Fig. 4) that we do not expect much change. The following was added to the text: "As the lateral melt already comprises a small fraction of the mass budget in the control run (Fig. 5), sensitivity runs with decreased lateral melting were not completed."*
*Similarly, Fig. 4 suggests than the 100x run already results in a majority (>50%) of the melt being partitioned to lateral melting. We do not think that a 1000x run is necessary to get the picture of sensitivity to this parameter and was beyond the scope of the runs completed here. The realism of the floe sizes and parameterization values inferred by these parameterizations is discussed in the response to comment 2-8, as "changes to the floe size are rather used as a catchall for factors impacting the sensitivity" (Sect. 2.2)* |
| 2-26 | P12 L287 is your SOB appropriate for the SO where MLD can be much larger than 10m | *The description of the SOM previously was not sufficiently clear; while the MLD has to be a minimum of 10 m, it is a depth prescribed based on an average from the fully coupled model, and is, in fact, often much deeper in the Southern Ocean. The rewritten text in section 2.1 makes this more clearer: "The prescribed MLD varies spatially based on climatological conditions simulated by CESM2.0 coupled simulations with an active ocean component. However, it is constant over time (i.e. doesn't vary seasonally or inter-annually), and has a minimum depth of 10 m."* |

| 2-27 | P14 L310 on what basis you state this? Clarify | *This statement has been removed "… indicate a larger potential role of lateral melting in the seasonal Antarctic ice pack") as we agree this was confusing here, and the point that we hoped to make was more clearly made elsewhere in this section.* |
|---|---|---|
| 2-28 | P16 L334 realistic as in consistent with observations? You have not discussed that much here at all. | *Yes, that was the intent here. The sentence has been revised to: "Though realistic values of OWFE are not able to be observationally determined…"* |

---

## Author Response (AR2)

We are grateful for the very helpful and kind reviews from both the reviewer and editor. Point-by-point responses to comments describing changes made in the revised manuscript are provided below in italics.

**Anonymous Referee #1**

**General Comments**

This is my second review of the manuscript, 'Arctic sea ice sensitivity to lateral melting representation in a coupled climate model' by Smith et al., (2021). In my first review, I suggested that this research provides valuable insight and discussion regarding the treatment of lateral melt in sea ice and climate models, highlighting in particular the use of variable lateral melt rate across thickness categories and the application of the concept of open water formation efficiency. This general assessment continues to hold for the updated manuscript.

I previously identified the following general points that I thought should be addressed before publication:

- The description of the methodology was insufficiently detailed.
- The need for more discussion of the limitations of the methodology and any conclusions reached.
- The conclusion reached regarding the role of ice-albedo feedback in changes to sea ice state was not justified by the methodology.
- The inclusion of further figures to add insight.
- The structure of discussion and conclusions.

In my opinion, the above points have been sufficiently addressed in the updated manuscript. I believe the manuscript is within the scope of The Cryosphere and merits publication. I have a few remaining comments and suggested edits that I will list below.

|     | Specific Comments                                                                                                                                                                                                                                                                                                                                                                                                                                                                                                                                                                 | Response                                                                                                                                                                                                                                                                                                                                                                                                                                                                                                                                                                                                                                                                                                                                                                                                             |
|-----|-----------------------------------------------------------------------------------------------------------------------------------------------------------------------------------------------------------------------------------------------------------------------------------------------------------------------------------------------------------------------------------------------------------------------------------------------------------------------------------------------------------------------------------------------------------------------------------|----------------------------------------------------------------------------------------------------------------------------------------------------------------------------------------------------------------------------------------------------------------------------------------------------------------------------------------------------------------------------------------------------------------------------------------------------------------------------------------------------------------------------------------------------------------------------------------------------------------------------------------------------------------------------------------------------------------------------------------------------------------------------------------------------------------------|
| 1-1 | P2 L37-44: In general, this section would
benefit from a more detailed explanation of the
points being made e.g., 'the representation of
the sea ice cover using an ice thickness
distribution results in a stronger albedo
feedback because of the impact on
thermodynamic processes'. Explain what the
impact is and how this interacts with the
albedo feedback.
In addition, the final comment, whilst valid,
does not clearly follow from the proceeding
section: 'and also the potential effect of lateral
melt on driving feedbacks.'. | The sentence referenced here has been edited
to read "because the better resolution of thin
ice enhances thermodynamic ice loss." We
choose to not further expand this section as
the introduction is already relatively lengthy.
The final sentence of the paragraph has been
re-written with the aim of clarifying what is
suggested by the Bitz et al (2001) study.
"Bitz et al. (2001) states that "resolving thin ice
[using the ice thickness distribution] eliminates
the need for partitioning an unrealistically high
fraction of heat flux toward lateral melt",
indicating the importance of the ice thickness
distribution in simulating melt rates. Lateral
melt can have an important role in driving
feedbacks in a manner similar to the
thickness." |
| 1-2 | P4 L124-125: It would be useful to explain why you use monthly averages for assessing                                                                                                                                                                                                                                                                                                                                                                                                                                                                                             | We have added explanation of why monthly averages are use: "for computation efficiency                                                                                                                                                                                                                                                                                                                                                                                                                                                                                                                                                                                                                                                                                                                               |
|     | changes to the mean sea ice state and daily                                                                                                                                                                                                                                                                                                                                                                                                                                                                                                                                       |                                                                                                                                                                                                                                                                                                                                                                                                                                                                                                                                                                                                                                                                                                                                                                                                                      |

|      | averages to examine the efficiency of melt                                                                                                                                                                                                                                                                                                                             | and for better comparison with other studies                                                                                                                                                                                                                                                                                                                                                                                                                         |
|------|------------------------------------------------------------------------------------------------------------------------------------------------------------------------------------------------------------------------------------------------------------------------------------------------------------------------------------------------------------------------|----------------------------------------------------------------------------------------------------------------------------------------------------------------------------------------------------------------------------------------------------------------------------------------------------------------------------------------------------------------------------------------------------------------------------------------------------------------------|
|      | processes.                                                                                                                                                                                                                                                                                                                                                             | (which typically use monthly averages)"                                                                                                                                                                                                                                                                                                                                                                                                                              |
| 1-3  | P12 L271-273: Did you use daily or monthly
model output for this analysis? Ideally it should
be daily to minimise averaging effects.                                                                                                                                                                                                                             | Yes, this analysis was completed using daily
model outputs averaged over the month, for
consistency with what we understood was
done in the Bateson et al., 2020 study.                                                                                                                                                                                                                                                                                     |
| 1-4  | P20 L443-445: Are you able to provide a reference for this statement regarding observations on the MOSAiC expedition?                                                                                                                                                                                                                                                  | Unfortunately given the timeline, we do not yet
have an appropriate citation for MOSAiC that
shows the relevant observations referred to,
but we have now included a refence to the Sea
Ice Overview paper (Nicolaus et al., 2021)
which includes brief mention of this data
collection.                                                                                                                                                           |
| 1-5  | Figure 4: Can you comment on the spatial
distribution of positive and negative changes in
sea ice concentration in panel (d), since this is a
fairly distinct pattern compared to panels (b)
and (c)?                                                                                                                                                      | The following sentence has been added to
Section 3.1: "The changes in spatial patterns
with the redistribution of lateral melt (Fig. 4d)
results from the patterns in ice thickness
distribution, where the relatively high
proportion of thick ice to thin ice in summer in
the Barents Sea results in an increase in
concentration."                                                                                                          |
| 1-6  | Figs 3 + 8: It would be helpful to briefly
comment in the text on how realistic the mean
sea ice state is for your reference simulation.                                                                                                                                                                                                                         | It is not possible to comment on how 'realistic'
sea ice is for pre-industrial control runs due to
the lack of comparison data for this forcing.
However, as we already state in the methods:
"Sea ice simulated by CESM2.0 over the
historical period has reasonable mean state
and variability in both hemispheres
(DeRepentigny et al., 2020)." This suggests
that the model is producing reasonable ice
based on the climate forcing. |
| 1-7  | Figs 5 + 10: The plots suggest melt volumes of
the order $10^{15} - 10^{16}$ m 3 , however the
difference between the minimum and
maximum sea ice volume shown in Fig. 3 is
about 1.5 x 10^13 m 3 i.e., the melt volumes
are too high. I am wondering if the axis is
supposed to have units of kg rather than m 3 . | Good catch! There was a unit error in Figures 5
and 10 that has now been remedied. This
correction has no impact on the results or
conclusions. Note that the melt/growth
volumes still appear somewhat high as there
are small amounts of growth and melt
throughout the year that are not reflected in
the volume change.                                                                                                                     |
|      | Technical corrections                                                                                                                                                                                                                                                                                                                                                  |                                                                                                                                                                                                                                                                                                                                                                                                                                                                      |
| 1-8  | P2 L34, P2 L49, P3 L61: Define acronyms or otherwise explain terms used.                                                                                                                                                                                                                                                                                               | All acronyms have been defined at first use or written out.                                                                                                                                                                                                                                                                                                                                                                                                          |
| 1-9  | P4 L108: Replace 'must' with 'much'.                                                                                                                                                                                                                                                                                                                                   | Done.                                                                                                                                                                                                                                                                                                                                                                                                                                                                |
| 1-10 | P4 L118: I am not sure what you mean by the final clause, 'included in prior versions'.                                                                                                                                                                                                                                                                                | This has been changed to "as standard in prior versions of the CICE model", which is hopefully more clear.                                                                                                                                                                                                                                                                                                                                                           |

| 1-11 | P6 L161: Can you double-check the definition              | Yes, good catch. We have changed the            |
|------|-----------------------------------------------------------|-------------------------------------------------|
|      | for $\Delta T$ (delta T)? In general, I would expect the  | beginning of the sentence to "The temperature   |
|      | inclusion of $\Delta$ (delta) to indicate the term refers | of the surface ocean above freezing " to |
|      | to a difference.                                          | indicate that the relevant temperature here is  |
|      |                                                           | the difference from freezing, hence the delta.  |

**Editor:**

**General Comments**

I thank the authors for submitting their revised manuscript to The Cryosphere. I especially appreciate the authors clear response to the comments raised in the previous review. Thank you.

**\* 37-44: Comment raised by reviewer:**

I suggest to BRIEFLY weave in the argument of "floe-size distribution" to address the reviewer's comment and to round up your background discussion. The paragraph 31-44 contains two major discussion points, which currently appear in conflict. Suggest to split them into separate paragraphs, or to join 31-37 with the previous paragraph.

The referenced sentences have been split into two paragraphs, which indeed better distinguishes the two distinct ideas. In the interest of not lengthening an already long introduction, we elect to not introduce any discussion of floe size distribution to this paragraph. The impact of floe size distribution on sea ice via the albedo feedback is already introduced later in the introduction. As this part of the introduction focuses on components of coupled climate models as they currently exist, it does not seem appropriate to introduce it here instead.

**\* Use of the term "lateral melt rate".**

The authors related the term "lateral melt rate" to the floe size (l65), and from then on equate changes in floe size to a linear change in "lateral melt rate". Pls include a short statement that this is a convenient approximation in the context of this study.

We do have a statement to this effect in Section 2.2: "As such, changes to the floe size are used as a catchall for factors impacting the sensitivity of the lateral melt rate." Acknowledging that this was a bit buried as presently presented, also added earlier in this paragraph "...where floe size is used as a convenient approximation for lateral melt rate in the context of this study."

|     | Technical corrections:                      | Response                                               |
|-----|---------------------------------------------|--------------------------------------------------------|
| E-1 | 123: Correct "1.0^°C to 1.5^°C" to "1.0C to | Done.                                                  |
|     | 1.5^∘C".                                    |                                                        |
| E-2 | 124: Add that you reuse EXISTING (??)       | The only existing run used is the "control"; all other |
|     | runs, and why you average over the last 25  | runs were completed for this study. In regard to       |
|     | years.                                      | why we have averaged over the last 25 years, we        |
|     |                                             | have added to the text "after the system has           |
|     |                                             | equilibrated and to minimize the contribution from     |
|     |                                             | internal variability."                                 |
| E-3 | Provide start and/or end year for the 25    | We add to the sentence referenced "(simulation         |
|     | years used.                                 | years 35-60)". Note that these are constant forcing    |
|     |                                             | runs so no calendar years are relevant.                |

| E-4  | 171: Correct "i.e." to "i.e.,".                 | Done.                                           |
|------|-------------------------------------------------|-------------------------------------------------|
| E-5  | 487: Rewrite "may be altered by" to "may        | Done.                                           |
|      | require revision due".                          |                                                 |
| E-6  | Fig.2, caption: Capitalize "effective" (y-axis) | Done.                                           |
|      | and "ice" (x-axis).                             |                                                 |
| E-7  | Fig.7: Move the legend to the right, so it      | Done.                                           |
|      | does NOT partially overlap with the figure      |                                                 |
|      | itself.                                         |                                                 |
| E-8  | Fig.10: Legend of symbols is missing            | Done.                                           |
|      | "dotted" for snow-ice. Pls add.                 |                                                 |
| E-9  | Fig.13, caption: For consistency with Fig.6,    | Done.                                           |
|      | replace "Black lines denote" with "Black        |                                                 |
|      | bars denote".                                   |                                                 |
| E-10 | Several figures: There are no x-axis labels     | Done. Appropriate label ('Date' or 'Month') has |
|      | when "Time" or unit is the x-axis, pls add      | been added for all figures.                     |
|      | for all relevant figures. E.g., Fig3 etc.       |                                                 |

---

## Author Response (AR3)

*Point-by-point responses are provided below in italics.*

Editor:
**Comments to the Author:**
I thank the authors for the quick turn around of their 2nd submitting their revision of their manuscript to The Cryosphere.

I am overall happy with their changes in response to the reviewer's and the editor's comments. Some minor errors have been detected, which should be fixed in the final version to be submitted by the author's into the editor's office/upload.

* Correct "Nicolas and et al" to "Nicolaus et al.", twice.
*Corrected.*

* In response to the "delta T" query (previous version l161), you replied that "above freezing" was added to the sentence on "The temperature of the surface ocean...", however in the latest version the complete sentence is missing. Suggest to add in, including the "above freezing".
*Very unsure how that sentence was deleted! Thank you very much for catching it, and it has now been added back in.*

* Acknowledgements: I did not see an acknowledgement to the reviewer's for their efforts to improve this paper. I suggest to add this.
*I am so sorry for this oversight! We are indeed grateful for the reviewers' and editor's comments, and have added a statement.*